# Graph Guided Diffusion: Unified Guidance for Conditional Graph Generation

## Abstract

Diffusion models have emerged as powerful generative models for graph generation, yet their use for conditional graph generation remains a fundamental challenge. In particular, guiding diffusion models on graphs with arbitrary reward signals is challenging: although gradient-based methods are powerful, they often fail because graph rewards are combinatorial and thus non-differentiable, complicating gradient-based guidance. We propose Graph Guided Diffusion (GGDiff), a novel guidance framework that interprets conditional diffusion on graphs as a stochastic control problem to address this challenge. GGDiff unifies multiple guidance strategies, including gradient-based guidance (for differentiable rewards), control-based guidance (using control signals from forward reward evaluations), and zeroth-order approximations (bridging gradient-based and gradient-free optimization). This comprehensive, plug-and-play framework enables zero-shot guidance of pre-trained diffusion models under both differentiable and non-differentiable reward functions, adapting well-established guidance techniques to graph generation. Our formulation balances computational efficiency, reward alignment, and sample quality, enabling practical conditional generation across diverse reward types. We demonstrate the efficacy of GGDiff in various tasks, including constraints on graph motifs, fairness, and link prediction, achieving superior alignment with target rewards while maintaining diversity and fidelity.

## 1 Introduction

Diffusion models have recently shown great promise for graph generation, enabling the synthesis of realistic graph structures across diverse domains such as drug design Yang et al. (2024), social networks Grover et al. (2019), and molecular dynamics Hoogeboom et al. (2022). A key motivation behind these models is their ability to serve as *flexible generative priors*, capturing complex dependencies in both graph topology and node features. However, most existing graph diffusion models focus on unconditional or controllable generation under simple objectives. Incorporating more *general forms of rewards or constraints*, such as enforcing specific structural properties, functional motifs, or domain-specific validity criteria like fairness, remains an essential and open challenge.

Recent advances in conditional graph generation typically modify the diffusion trajectory using a conditional gradient to steer the process toward sampling from the desired conditional distribution. DiGress Vignac et al. (2023) combines a learnable regressor with classifier guidance Ho & Salimans (2021), while LGD Zhou et al. (2024) adopts a similar gradient-based strategy in a latent space instead of a discrete domain like DiGress. However, these approaches require differentiable constraints, which limits their applicability in more complex graph generation tasks where constraints might be black-box functions without tractable gradients or involve discrete structures. Closer to our approach, PRODIGY Sharma et al. (2024) enforces hard constraints through projected sampling along the diffusion trajectory using the bisection method Boyd & Vandenberghe (2004). This requires closed-form projection operators for a time-efficient implementation, which are often unavailable for complex constraints, necessitating expensive solvers, and limiting the applicability to more sophisticated conditions. As a result, no existing method can flexibly and effectively handle arbitrary, non-differentiable, or complex constraints in the sampling process for conditional graph generation.

In this work, we propose Graph Guided Diffusion (GGDiff), a general guidance framework for graph generation that interprets conditional graph generation as a stochastic optimal control (SOC)

Figure 1: Illustration of GGDiff, a method that guides the generation of graphs to satisfy a set of constraints (in this case, the constraint is fairness). The guidance $\mathbf{U}_t$ is a local direction obtained via SOC, and approximated using ZO techniques, like the multi-point estimate shown here.

problem. By casting the task as a control problem, we reformulate guided diffusion as a conditional generation process with an additional control variable defined as a linear drift term. Inspired by recent advances in SOC for diffusion models Pandey et al. (2025); Huang et al. (2024); Rout et al. (2025), we optimize this control via path integral control, which provides an analytical yet intractable gradient. To overcome this limitation, we introduce gradient-free approximations based on zeroth-order (ZO) optimization techniques Liu et al. (2020; 2018). Our formulation generalizes several gradient-free strategies introduced previously Huang et al. (2024) and offers new possibilities.

GGDiff unifies various existing guidance methods in a single framework and is plug-and-play, allowing zero-shot guidance of pre-trained diffusion models under both differentiable and non-differentiable reward functions. We validate the advantages of GGDiff through extensive experiments on a wide range of constraints, including structural constraints, fairness, and link prediction. Our results demonstrate GGDiff's versatility in guiding graph generation not only towards constraints previously explored in the literature (beating current state-of-the-art architectures) but also towards arbitrary, user-defined desired outcomes (such as fair or incomplete graphs), effectively balancing precise outcome satisfaction with the preservation of the underlying graph family's characteristics.

To summarize, our contributions are threefold:

- We propose GGDiff, a framework for conditional graph generation that handles both differentiable and non-differentiable rewards by reformulating the problem as a stochastic optimal control (SOC) task.
- We introduce a general gradient-free ZO optimization formulation to handle non-differentiable rewards, enabling optimization without requiring tractable gradients.
- We conduct extensive experiments on structural, fairness, and link prediction constraints, demonstrating GGDiff's superior performance and flexibility over existing methods.

## 2 BACKGROUND AND RELATED WORKS

We review graph diffusion models in the continuous domain in Section 2.1, and then explain how they can be used in the context of inverse problems in Section 2.2.

### 2.1 DIFFUSION MODELS ON GRAPHS

Diffusion models (Sohl-Dickstein et al., 2015; Ho et al., 2020; Song et al., 2021) are composed of two processes: i) a forward process that starts with clean data and gradually adds noise; and ii) a reverse process that learns to generate new data by iteratively denoising its diffused version. While graph diffusion models have been developed in both *continuous* Niu et al. (2020); Jo et al. (2022) and *discrete* domains Vignac et al. (2023); Chen et al. (2023), in this work we focus on the continuous case. We represent a graph as $\mathbf{G}_0 = \{\mathbf{X}_0, \mathbf{A}_0\}$, where $\mathbf{X}_0 \in \mathbb{R}^{N \times F}$ are node features and $\mathbf{A}_0 \in \mathbb{R}^{N \times N}$ is the weighted adjacency matrix. Then, following GDSS (Jo et al., 2022), the forward diffusion process is defined by the stochastic differential equation $d\mathbf{G}_t = -\frac{1}{2}\beta(t)\mathbf{G}_t\, dt + \sqrt{\beta(t)}\, d\mathbf{W}_t, \quad t \in [0, T]$, where $\beta(t)$ controls the noise schedule and is given by $\beta(t) := \beta_{\min} + (\beta_{\max} - \beta_{\min})\frac{t}{T}$. Here, $\mathbf{W}_t$ denotes standard Brownian motion. This process is designed such that the distribution of $\mathbf{G}_T$

converges to a standard Gaussian as $t \to T$. Notice that by construction, the forward process allows for interaction between the adjacency matrix and the node features. Based on this forward process, we define the reverse process as $d\mathbf{G}_t = [-\frac{1}{2}\mathbf{G}_t - g(t)^2 \nabla_{\mathbf{G}_t} \log p(\mathbf{G}_t)]dt + g(t)d\mathbf{W}_t$, where $\nabla_{\mathbf{G}_t} \log p(\mathbf{G}_t)$ is the *score function*, which is unknown, and $g(t) = \sqrt{\beta(t)}$. In particular, GDSS considers two different score functions, namely $\nabla_{\mathbf{A}_t} \log p(\mathbf{A}_t)$ and $\nabla_{\mathbf{X}_t} \log p(\mathbf{X}_t)$.

Notice that the requirement for sampling, i.e., running the reverse process, is to have access to the score functions $\nabla_{\mathbf{A}_t} \log p(\mathbf{A}_t)$ and $\nabla_{\mathbf{X}_t} \log p(\mathbf{X}_t)$, which are generally unknown. Instead, we approximate them with *score networks* $\epsilon_{\boldsymbol{\theta}_A}(\mathbf{A}_t, t) \approx -\sigma_t \nabla_{\mathbf{A}_t} \log p(\mathbf{A}_t)$ and $\epsilon_{\boldsymbol{\theta}_X}(\mathbf{X}_t, t) \approx -\sigma_t \nabla_{\mathbf{X}_t} \log p(\mathbf{X}_t)$, and learn them by minimizing the denoising score-matching loss (Vincent, 2011). After training, samples are generated using samplers like DDPM (Ho et al., 2020) and DDIM (Song et al., 2020).

## 2.2 Controllable Generation of Graphs With Continuous Diffusion Models

Given a condition $\mathcal{C}$ and a reward function $r(\mathbf{G}_0)$ that quantifies how close the sample $\mathbf{G}_0$ is to meeting $\mathcal{C}$, our objective is to generate graphs $\mathbf{G}_0$ that maximize the reward function. From a Bayesian perspective, this problem boils down to sampling from the posterior $p(\mathbf{G}_0|\mathcal{C}) \propto p(\mathcal{C}|\mathbf{G}_0)p(\mathbf{G}_0)$ where $p(\mathcal{C}|\mathbf{G}_0) \propto \exp\left(r(\mathbf{G}_0)\right)$ is a likelihood term and $p(\mathbf{G}_0)$ is a prior given by the pre-trained diffusion model. We now describe previous works for both differentiable and non-differentiable reward functions.

**Controllable generation with differentiable rewards.** A plethora of works have proposed solutions for solving inverse problems in images (Chung et al., 2022; Mardani et al., 2024; Zilberstein et al., 2024; 2025). In general, these works assume a differentiable reward – the likelihood associated with a noisy measurement–, which allows us to compute the *conditional score* at noise level $t$ obtained via Bayes' rule

$$\nabla_{\mathbf{G}_t} \log p(\mathbf{G}_t|\mathcal{C}) = \nabla_{\mathbf{G}_t} p(\mathcal{C}|\mathbf{G}_t) + \nabla_{\mathbf{G}_t} \log p(\mathbf{G}_t). \tag{1}$$

This naturally allows us to incorporate the diffusion model as the prior. However, the score associated with the likelihood term is intractable, as seen from $p(\mathcal{C}|\mathbf{G}_t) = \int p(\mathcal{C}|\mathbf{G}_0)p(\mathbf{G}_0|\mathbf{G}_t)d\mathbf{G}_0$. To circumvent this, different works proposed to use Gaussian approximation of $p(\mathbf{G}_0|\mathbf{G}_t)$ centered at the minimum-mean squared error (MMSE) denoiser, which can be computed using Tweedie's formula $\mathbb{E}[\mathbf{G}_0|\mathbf{G}_t] = \frac{1}{\alpha_t}\left(\mathbf{G}_t + \sigma_t^2 \nabla_{\mathbf{G}_t} \log p(\mathbf{G}_t, t)\right)$. In the context of graphs, DiGress Vignac et al. (2023) and LGD Zhou et al. (2024) incorporates a guidance via a learned model (similar to classifier-free guidance), requiring an extra learnable model. However, this approximation remains largely unexplored in the context of graph inverse problems, mainly because it requires (and assumes) *differentiable* rewards, which in general are not available for graph generation (You et al., 2018).

**Controllable generation with non-differentiable rewards.** Recently, PRODIGY Sharma et al. (2024) explored an alternative for controllable graph generation with non-differentiable constraints. Their core idea involves an unconditional generation step from $\mathbf{G}_t$ to produce a candidate $\hat{\mathbf{G}}_{t-1}$, followed by a projection step. Formally, after obtaining $\hat{\mathbf{G}}_{t-1}$, they apply a projection operator $\Pi_{\mathcal{C}}(\hat{\mathbf{G}}_{t-1}) = \text{argmin}_{\mathbf{Z} \in \mathcal{C}}||\mathbf{Z} - \hat{\mathbf{G}}_{t-1}||_2^2$. While direct projection ensures the sample satisfies $\mathcal{C}$, applying it fully at each noise level can disrupt the learned reverse trajectory's smoothness. To address this, PRODIGY proposes a partial update: $\mathbf{G}_{t-1} \leftarrow (1 - \gamma_t)\hat{\mathbf{G}}_{t-1} + \gamma_t \Pi_{\mathcal{C}}\left(\hat{\mathbf{G}}_{t-1}\right)$, where $\gamma_t$ balances constraint adherence with the original diffusion path. Although PRODIGY implements this constrained step efficiently for simple constraints with closed-form expression for $\Pi_{\mathcal{C}}(.)$, its reliance on such operators limits applicability to general reward functions without incurring prohibitive runtime. Furthermore, applying the projection operator directly to the noisy variable $\mathbf{G}_t$ rather than the denoised estimate $\mathbb{E}[\mathbf{G}_0|\mathbf{G}_t]$ is misaligned with the reward domain, which is defined at the data level ($t = 0$). In addition, the work in Madeira et al. (2024) considered more complex constraints. In a nutshell, they combine projection operators combined with an edge-absorbing model. While effective, this method is computationally demanding due to its combinatorial nature and for using discrete diffusion models.

## 3  CONTROLLABLE GENERATION OF GRAPHS WITH GENERAL REWARDS

In Section 3.1, we formulate the generation of graph conditionals as a SOC problem. Then, in Section 3.2 we propose different approximate solutions to design the control for conditional graph generation: first, in Section 3.2.1 we introduce our approximation for differentiable rewards; second, in Section 3.2.2 we propose a ZO approximation, which unifies several existing guidance policies for non-differentiable rewards.

### 3.1  CONDITIONAL GENERATION: A SOC APPROACH

The goal of our method is to *steer* a pre-trained diffusion model to sample from the posterior distribution. Importantly, we seek an algorithm that can handle general reward functions, even non-differentiable ones. To tackle this, we propose to leverage SOC Van Handel (2007). In particular, given an uncontrolled diffusion process $\mathcal{Q}$ (defined in Section 2.1), we define a controlled one $\mathcal{Q}^{\mathcal{C}}$ given by

$$\mathcal{Q}^{\mathcal{C}} : d\mathbf{G}_t^{\mathcal{C}} = \left[ -\frac{1}{2}\mathbf{G}_t^{\mathcal{C}} - g(t)^2 \nabla_{\mathbf{G}_t^{\mathcal{C}}} \log p(\mathbf{G}_t^{\mathcal{C}}) + g(t)\mathbf{U}(\mathbf{G}_t^{\mathcal{C}}, t) \right] dt + g(t)d\mathbf{W}_t, \quad t \in [T, 0]. \quad (2)$$

Thus, the goal is to design the control $\{\mathbf{U}(\mathbf{G}_t^{\mathcal{C}}, t)\}_{t \in [0,T]}$ to modify the trajectory of the controlled process $\mathcal{Q}^{\mathcal{C}}$ such that the generated samples belong to the target distribution. We formalize this as a SOC problem, where we solve the following optimization problem

$$\min_{\mathbf{U} \in \mathcal{U}} \mathbb{E}\left[ \int_0^T \lambda \frac{\|\mathbf{U}\left(\mathbf{G}_t^{\mathcal{C}}, t\right)\|_F^2}{2} dt - r\left(\mathbf{G}_0^{\mathcal{C}}\right) \right] \quad \text{s.t. } \mathcal{Q}^{\mathcal{C}}. \quad (3)$$

The terminal cost in (3) represents a desired constraint for the final state $\mathbf{G}_0$ quantified by the reward $r(.)$, which is maximized (thus, the negative sign), while the transient term is a regularization term that penalizes large deviation from the uncontrolled process by promoting the energy of the controller in (2) to be small. The solution of (3) is given by the Feynman-Kac formula, a well-known result from the optimal control theory Pavon (1989), given by

$$\mathbf{U}^*(\mathbf{G}_t^{\mathcal{C}}, t) = -g(t)\nabla_{\mathbf{G}_t^{\mathcal{C}}} \log \mathbb{E}_{p^{\text{pre}}}\left[ \exp\left( \frac{-r(\mathbf{G}_0^{\mathcal{C}})}{\lambda} \right) \Big| \mathbf{G}_t^{\mathcal{C}} \right], \quad (4)$$

where $p^{\text{pre}}$ refers to the unconditional process, which in our case is the pre-trained diffusion model. The solution in (4) is obtained as the solution of the *linear* version of the Hamilton-Jacobi-Bellman (HJB) equation Evans (2022), obtained after the exponential transformation; we refer to Appendix A for more details on the derivation.

Given the optimal control, we now focus on how to implement it.

### 3.2  ESTIMATION OF THE OPTIMAL CONTROL: A GREEDY SOLUTION

Although the expression for the optimal control derived from the Feynman-Kac formula (4) is theoretically exact, its direct computation is often intractable. Evaluating the expectation and its gradient would require simulating numerous trajectories of the uncontrolled process from the current state $\mathbf{G}_t^{\mathcal{C}}$ to the final state $\mathbf{G}_0^{\mathcal{C}}$ at each step of the generation process to estimate $p^{\text{pre}}$, and then backpropagating through the diffusion trajectory. This is computationally prohibitive.

We resort to a *greedy* approximation strategy to overcome this. This approach simplifies the problem by approximating the complex gradient of the log-expectation term in (4) using primarily the current state information $\mathbf{G}_t^{\mathcal{C}}$ and a one-step estimate of the clean sample $\hat{\mathbf{G}}_0^{\mathcal{C}}$. Such an approximation implies that the control decision at time $t$ does not fully account for the entire future trajectory, potentially leading to suboptimal choices, especially in the early stages of the reverse diffusion process. However, the impact of such approximation errors may often diminish as $t \to 0$ and the state $\mathbf{G}_t^{\mathcal{C}}$ gets closer to the data. We now detail this approximation for the cases of $(i)$ differentiable rewards and $(ii)$ non-differentiable counterparts.

### 3.2.1 DIFFERENTIABLE REWARDS

When the reward function $r(\cdot)$ is differentiable, we can derive a tractable approximation for the optimal control $\mathbf{U}^*(\mathbf{G}_t^{\mathcal{C}}, t)$. The primary challenge lies in evaluating the gradient of the log-expectation term. To circumvent this, we use Tweedie's formula (see Section 2.2) to compute the MMSE denoiser $\mathbb{E}[\mathbf{G}_0^{\mathcal{C}}|\mathbf{G}_t^{\mathcal{C}}] = \hat{\mathbf{G}}_0^{\mathcal{C}}(\mathbf{G}_t^{\mathcal{C}})$ and approximate the conditional expectation in (4) as

$$\mathbb{E}_{p^{\text{pre}}}\left[\exp\left(\frac{-r(\mathbf{G}_0^{\mathcal{C}})}{\lambda}\right)\Big|\mathbf{G}_t^{\mathcal{C}}\right] \approx \exp\left(\frac{-r(\hat{\mathbf{G}}_0^{\mathcal{C}}(\mathbf{G}_t^{\mathcal{C}}))}{\lambda}\right), \tag{5}$$

where the underlying assumption is that $p(\mathbf{G}_0^{\mathcal{C}}|\mathbf{G}_t^{\mathcal{C}}) = \delta(\mathbf{G}_0^{\mathcal{C}} - \hat{\mathbf{G}}_0^{\mathcal{C}}(\mathbf{G}_t^{\mathcal{C}}))$ with $\delta(.)$ denoting a Dirac delta function. This approximation becomes increasingly better as $t \to 0$ (i.e., towards the end of the reverse diffusion process), as $\hat{\mathbf{G}}_0^{\mathcal{C}}(\mathbf{G}_t^{\mathcal{C}})$ becomes a better estimate of $\mathbf{G}_0^{\mathcal{C}}$.

Substituting this approximation into the exact optimal control formula in (4) leads to

$$\mathbf{U}^*(\mathbf{G}_t^{\mathcal{C}}, t) \approx \frac{g(t)}{\lambda}\nabla_{\mathbf{G}_t^{\mathcal{C}}}r(\hat{\mathbf{G}}_0^{\mathcal{C}}(\mathbf{G}_t^{\mathcal{C}})). \tag{6}$$

This final expression provides a tractable, greedy approximation for the optimal control. The control term now directly involves the gradient of the reward function $r(\cdot)$ evaluated at the one-step denoised estimate $\hat{\mathbf{G}}_0^{\mathcal{C}}$. The term $1/\lambda$ acts as a scaling factor for the guidance. This formulation resembles guidance techniques in diffusion models, as observed by Huang et al. (2024); Uehara et al. (2025). For example, if the reward $r(\mathbf{G}_0)$ is proportional to the log-likelihood of a condition $\mathcal{C}$, that is, $r(\mathbf{G}_0) \propto -\log p(\mathcal{C}|\mathbf{G}_0)$, then the optimal controls boils down to the DPS approximation Chung et al. (2022).

### 3.2.2 NON-DIFFERENTIABLE REWARDS

In many practical scenarios of controlled graph generation, the reward function $r(\cdot)$ is non-differentiable with respect to the generated graph $\hat{\mathbf{G}}_0^{\mathcal{C}}$, rendering gradient-based approximations like (6) intractable.

To address this, we propose to determine the control input $\mathbf{U}(\mathbf{G}_t, t)$ using an approach inspired by gradient-free optimization methods Larson et al. (2019) and ZO optimization Liu et al. (2020). The objective at each time $t$ is to find a control $\mathbf{U}(\mathbf{G}_t, t)$ that steers the diffusion trajectory towards graphs yielding a high reward $r(\mathbf{G}_0^{\mathcal{C}})$. Similar to the differentiable case, we use Tweedie's formula to compute a one-step denoised version of the final graph to evaluate the reward at each time step. Given this approximation, we formally seek to find a direction $\mathbf{U}_t^*$

$$\mathbf{U}_t^* = \underset{\mathbf{U}_t}{\arg\max}\; r\left(\hat{\mathbf{G}}_0^{\mathcal{C}}(\mathbf{G}_t^{\mathcal{C}} + \mu\mathbf{U}_t)\right), \tag{7}$$

Here, $\mathbf{G}_t^{\mathcal{C}} + \mu\mathbf{U}_t$ denotes the perturbed version of $\mathbf{G}_t^{\mathcal{C}}$, which is the generated graph with the reference model at time $t$ (before applying the guidance) following the control direction $\mathbf{U}_t$, and $\mu$ is a smoothing parameter (which depends on the noise schedule of the diffusion process). To find $\mathbf{U}_t^*$, we define a general ZO estimator for the gradient of the reward that depends on evaluations of $r(.)$ as

$$\hat{\nabla}r(\mathbf{G}_t^{\mathcal{C}}) := \mathbb{E}_{\mathbf{U}_t \sim \mathcal{D}}\left[w(\mathbf{U}_t)\, r\left(\hat{\mathbf{G}}_0^{\mathcal{C}}(\mathbf{G}_t^{\mathcal{C}} + \mu\mathbf{U}_t)\right) \cdot \mathbf{U}_t\right], \tag{8}$$

where $\mathcal{D}$ is a distribution over directions (typically Gaussian) and $w(\mathbf{U}_t)$ is a direction-dependent weighting function. Notably, this formulation unifies several previous gradient-free estimators. However, it is important to remark that traditional ZO optimization assumes the objective is differentiable but the gradient is inaccessible. In contrast, in our setting the reward function $r(.)$ is inherently non-differentiable, often defined via a discrete or combinatorial metric over generated graphs. Nevertheless, we treat the reward as a black-box function and employ randomized directional evaluations to define a *pseudo-gradient* direction that can guide the controlled process. Thus, the ZO estimator in (8) should be interpreted as a surrogate direction that correlates with improvements in the reward, rather than an unbiased estimator of a true gradient (see Appendix B.1). We now present three practical ZO estimators that instantiate (8).

---

**Algorithm 1** GGDiff for controllable generation on graphs

---

**Require:** $T, \epsilon_{\boldsymbol{\theta}}(\mathbf{G}_t, t), N, k, \mu, \{\alpha_t\}_{t=0}^T, \{\sigma_t\}_{t=0}^T, r(\cdot)$
 1: Sample $\mathbf{G}_T^{\mathcal{C}}$ from $p(\mathbf{G}_T)$.
 2: **for** $t = T - 1$ to $1$ **do**
 3: $\quad \mathbf{G}_t^{\mathcal{C}} = \frac{1}{\sqrt{\alpha_{t+1}}} \left( \mathbf{G}_{t+1}^{\mathcal{C}} - \frac{1-\alpha_{t+1}}{\sqrt{1-\bar{\alpha}_{t+1}}} \epsilon_{\boldsymbol{\theta}}(\mathbf{G}_{t+1}^{\mathcal{C}}, t+1) \right)$ (DDPM update).
 4: $\quad$ **if** $r$ is differentiable **then**
 5: $\qquad$ Compute $\hat{\mathbf{G}}_0^{\mathcal{C}}(\mathbf{G}_t^{\mathcal{C}}) = \frac{1}{\alpha_t} \left( \mathbf{G}_t^{\mathcal{C}} + \sigma_t^2 \epsilon_{\boldsymbol{\theta}}(\mathbf{G}_t^{\mathcal{C}}, t) \right)$.
 6: $\qquad$ Compute $\mathbf{U}_t = \nabla_{\mathbf{G}_t^{\mathcal{C}}} r(\hat{\mathbf{G}}_0^{\mathcal{C}}(\mathbf{G}_t^{\mathcal{C}}))$ using (6).
 7: $\quad$ **else**
 8: $\qquad$ Sample $N$ candidates $\{\mathbf{U}_t^{(1)}, \ldots, \mathbf{U}_t^{(N)}\} \sim \mathcal{N}(\mathbf{0}, \mathbf{I})$.
 9: $\qquad$ Compute $\tilde{\mathbf{G}}_t^{\mathcal{C},(i)} = \mathbf{G}_t^{\mathcal{C}} + k\mathbf{U}_t^{(i)}$ for $i = 1, \cdots, N$.
10: $\qquad$ Compute $\hat{\mathbf{G}}_0^{\mathcal{C},(i)} = \frac{1}{\alpha_t} \left( \tilde{\mathbf{G}}_t^{\mathcal{C},(i)} + \sigma_t^2 \epsilon_{\boldsymbol{\theta}}(\tilde{\mathbf{G}}_t^{\mathcal{C},(i)}, t) \right)$ for $i = 1, \cdots, N$.
11: $\qquad$ Approximate $\hat{\nabla} r(\mathbf{G}_t^{\mathcal{C}})$ using (10), (11), (12)
12: $\qquad$ **if** Approximation of $\hat{\nabla} r(\mathbf{G}_t^{\mathcal{C}})$ is (11) **then**
13: $\qquad\quad$ Set $\mathbf{U}_t = \operatorname{argmax}_{\mathbf{U}_t^{(i)}} r(\hat{\mathbf{G}}_0^{\mathcal{C},(i)})$.
14: $\qquad$ **else if** Approximation of $\hat{\nabla} r(\mathbf{G}_t^{\mathcal{C}})$ is (10) or (12) **then**
15: $\qquad\quad$ Set $\mathbf{U}_t = \hat{\nabla} r(\mathbf{G}_t^{\mathcal{C}})$.
16: $\qquad$ **end if**
17: $\quad$ **end if**
18: $\quad \mathbf{G}_t^{\mathcal{C}} = \mathbf{G}_t^{\mathcal{C}} + k\mathbf{U}_t$.
19: **end for**
20: **return** $\mathbf{G}_0^{\mathcal{C}}$

---

**One-point (and two-point) gradient estimators.** The one-point estimator samples a single perturbation direction $\mathbf{U}_t \sim \mathcal{N}(\mathbf{0}, \mathbf{I})$ and evaluates the reward by perturbing the unconditional generated graph with this single direction. The estimated gradient is given by

$$\hat{\nabla} r(\mathbf{G}_t^{\mathcal{C}}) = \frac{\phi(d)}{\mu} r \left( \hat{\mathbf{G}}_0^{\mathcal{C}}(\mathbf{G}_t^{\mathcal{C}} + \mu \mathbf{U}_t) \right) \cdot \mathbf{U}_t, \tag{9}$$

where $\phi(d)$ is a scaling factor that depends on $\mathcal{D}$; for $\mathcal{D}$ Gaussian, we have $\phi(d) = 1$. This control corresponds to $w(\mathbf{U}_t) = \frac{\phi(d)}{\mu}$. In classical ZO, this estimator is an unbiased estimator of the smoothed version of $r(.)$ over a random perturbation, i.e., $\mathbb{E}_{\mathbf{U}_t \sim \mathcal{D}}[r(\hat{\mathbf{G}}_0^{\mathcal{C}}(\mathbf{G}_t^{\mathcal{C}} + \mu \mathbf{U}_t))]$, but a *biased* estimator of the true reward gradient (when $\mu = 0$) and has high variance (the variance explodes as $\mu$ increases to 0) Berahas et al. (2022). To eliminate this problem, we can use instead a two-point gradient estimator given by

$$\hat{\nabla} r(\mathbf{G}_t^{\mathcal{C}}) = \frac{\phi(d)}{\mu} \left[ r \left( \hat{\mathbf{G}}_0^{\mathcal{C}}(\mathbf{G}_t^{\mathcal{C}} + \mu \mathbf{U}_t) \right) - r \left( \hat{\mathbf{G}}_0^{\mathcal{C}}(\mathbf{G}_t^{\mathcal{C}}) \right) \right] \cdot \mathbf{U}_t, \tag{10}$$

which is used in practice in general. For cases where $r(.)$ is differentiable, the estimator in (10) is unbiased w.r.t. true gradient (under the assumption that $\mathbb{E}_{\mathbf{U}_t \sim \mathcal{D}}[\mathbf{U}_t] = 0$ and when $\mu \to 0$).

**Best-of-$N$ direction (greedy ZO).** Instead of sampling a single direction, this method samples $N$ candidate directions $\{\mathbf{U}_t^{(1)}, \ldots, \mathbf{U}_t^{(N)}\} \sim \mathcal{N}(\mathbf{0}, \mathbf{I})$, and chooses the one that maximizes the reward after denoising:

$$\mathbf{U}_t^{(i)} = \operatorname*{argmax}_{\{\mathbf{U}_t^{(1)}, \ldots, \mathbf{U}_t^{(N)}\}} r \left( \hat{\mathbf{G}}_0^{\mathcal{C}}(\mathbf{G}_t^{\mathcal{C}} + \mu \mathbf{U}_t) \right) \cdot \mathbf{U}_t. \tag{11}$$

The final control is then set as $\mathbf{U}_t = k \cdot \mathbf{U}_t^{(i)}$, where $k$ is a step size or scaling factor. This corresponds to using $w(\mathbf{U}_t) = \mathbb{1}(\mathbf{U}_t = \mathbf{U}_t^{(i)})$ in (8), where $\mathbb{1}$ represents the indicator function. While this method introduces bias, it often leads to effective and low-variance updates, especially when $r(\cdot)$ is highly non-smooth or sparse.

Table 1: Comparison of metrics across datasets and constraints.

| Constraint | Method | Ego Small | | Community Small | | Enzymes | |
|---|---|---|---|---|---|---|---|
| | | $\Delta$ MMD $\uparrow$ | Val$_\mathcal{C}$ $\uparrow$ | $\Delta$ MMD $\uparrow$ | Val$_\mathcal{C}$ $\uparrow$ | $\Delta$ MMD $\uparrow$ | Val$_\mathcal{C}$ $\uparrow$ |
| Max Degree | GGDiff-G | 0.11 | 0.87 | -0.54 | 0.95 | -0.37 | 0.98 |
| | GGDiff-C | **0.15** | **0.90** | -0.73 | **1.00** | -0.39 | **1.00** |
| | GGDiff-Z | 0.08 | 0.86 | -0.26 | 0.78 | -0.36 | 0.89 |
| | PRODIGY | 0.09 | 0.64 | **-0.16** | 0.98 | **0.07** | 0.95 |
| | Uncons. | 0.00 | 0.33 | 0.00 | 0.42 | 0.00 | 0.08 |
| Edge Count | GGDiff-G | -0.07 | **0.91** | -0.33 | 0.84 | -0.47 | **1.00** |
| | GGDiff-C | 0.27 | 0.63 | **-0.17** | 0.91 | -0.29 | 0.94 |
| | GGDiff-Z | **0.28** | 0.67 | -0.38 | 0.73 | -0.12 | 0.69 |
| | PRODIGY | 0.27 | 0.70 | -0.39 | **1.00** | **-0.10** | **1.00** |
| | Uncons. | 0.00 | 0.16 | 0.00 | 0.20 | 0.00 | 0.09 |
| Triangle Count | GGDiff-G | **0.03** | **0.96** | -0.31 | 0.95 | -0.03 | 0.98 |
| | GGDiff-C | 0.01 | 0.89 | -1.00 | **1.00** | -0.01 | **1.00** |
| | GGDiff-Z | -0.07 | 0.88 | -0.14 | 0.85 | -0.04 | **1.00** |
| | PRODIGY | -0.01 | 0.52 | **-0.13** | 0.72 | **0.17** | 0.94 |
| | Uncons. | 0.00 | 0.62 | 0.00 | 0.19 | 0.00 | 0.50 |

**Multi-point gradient estimator (averaged random search).** This variant also samples $N$ directions $\{\mathbf{U}_t^{(1)}, \ldots, \mathbf{U}_t^{(N)}\} \sim \mathcal{N}(\mathbf{0}, \mathbf{I})$, but instead of selecting the best, it forms a weighted average of all directions using their corresponding reward evaluations

$$\hat{\nabla} r(\mathbf{G}_t^\mathcal{C}) = \frac{1}{N\mu} \sum_{i=1}^{N} \left[ r \left( \hat{\mathbf{G}}_0^\mathcal{C}(\mathbf{G}_t^\mathcal{C} + \mu \mathbf{U}_t^{(i)}) \right) - r \left( \hat{\mathbf{G}}_0^\mathcal{C}(\mathbf{G}_t^\mathcal{C}) \right) \right] \cdot \mathbf{U}_t^{(i)}. \tag{12}$$

This approach reduces variance compared to both one-point and two-point estimators while maintaining approximate unbiasedness. It is especially useful when the reward landscape is moderately smooth, enabling the use of reward information from all sampled directions. We defer a quantitative analysis of variance and performance of the three estimators to Appendix B. Overall, these estimators offer flexible trade-offs between estimator quality and query complexity. In our setting, we find that the best-of-$N$ direction yields superior performance in discrete and non-differentiable environments, typical of graph-based objectives.

**Final algorithm.** We put everything together and show our proposed algorithm in Alg. 1.

## 4 EXPERIMENTS

We evaluate the efficacy of our GGDiff framework for conditional graph generation across various tasks and reward functions. We compare the performance of the proposed guidance strategies within the GGDiff framework: "GGDiff-G", which is the gradient-based guidance framework for differentiable rewards; "GGDiff-C", which implements the greedy Best-of-$N$ approach for non-differentiable rewards; and "GGDiff-Z", which employs the multi-point gradient estimator. We use GDSS (Jo et al., 2022) as the underlying diffusion model. As baselines, we also report metrics for PRODIGY (Sharma et al., 2024), a state-of-the-art method for conditional graph generation, the guidance framework of DiGress in Vignac et al. (2023, Sec. 5), and results from unconstrained graph generation to highlight the impact of guidance.

Our experiments cover three main areas: constrained graph generation (Section 4.1), where we assess the ability of GGDiff to generate graphs satisfying specific structural properties; fair graph generation (Section 4.2), evaluating the framework's performance in generating graphs that adhere to fairness criteria; and link prediction (Section 4.3), where we aim to generate graphs consistent with partially observed adjacency matrices. The performance of PRODIGY will only be assessed in the first set of experiments (Section 4.1), as it is unable to handle more complex guidance scenarios like the ones

presented in Sections 4.2 and 4.3. Further experimental details, along with more experiments, can be found in Appendix E.

## 4.1 Constrained Graph Generation

We first evaluate GGDiff's performance on constrained graph generation tasks, replicating the experimental setup from the PRODIGY paper (Sharma et al., 2024) to enable direct comparison. For this set of experiments, we impose constraints on the maximum degree, edge count, and maximum number of triangles of the generated graphs, on the ego small, community small, and enzymes datasets, described in Appendix E. To evaluate performance, we use two key metrics

Table 2: Cycle minimization.

| Method | # Cycles | % Valid Egonet |
|---|---|---|
| GGDiff-C | $0.31 \pm 0.53$ | 91.00 |
| GGDiff-Z | $2.64 \pm 7.28$ | 98.00 |
| Uncons. | $9.52 \pm 40.5$ | 100.00 |

rics proposed in Sharma et al. (2024). The first is used to assess closeness to the original graph distribution, $\Delta$ MMD, which measures the difference between the MMD values of the unconstrained dataset and the constrained generated graphs, i.e., $\Delta$ MMD := MMD $(\{\mathbf{G}_{0,i}\}_i, \{\mathbf{G}_i^{te}\}_i) -$ MMD $(\{\mathbf{G}_{0,i}^{\mathcal{C}}\}_i, \{\mathbf{G}_i^{te}\}_i)$, where $\mathbf{G}_i^{te}$ are the graphs in the test dataset (higher values of $\Delta$ MMD indicate that the generated graphs are closer to the original data distribution). The second metric is used to assess constraint satisfaction, $\text{Val}_{\mathcal{C}}$, representing the fraction of generated graphs that successfully fulfill the imposed constraint.

The results for this set of experiments are presented in Table 1. They demonstrate that our GGDiff methods generally achieve superior performance compared to baselines. Specifically, GGDiff variants tend to exhibit higher $\Delta$ MMD values while also showing higher $\text{Val}_{\mathcal{C}}$ scores, demonstrating their capability to satisfy structural constraints without deviating significantly from the prior distribution of the datasets; we defer to Appendix E.3.1 for more details about which variant to choose.

Beyond the constraints considered in Sharma et al. (2024), we evaluate our framework on the task of minimizing cycles and generating star graphs (Appendix E.3.2), illustrating its ability to optimize arbitrary non-differentiable rewards. Since cycle counting is combinatorially complex, only our ZO-based methods apply. As shown in Table 2, both guidance strategies we introduce successfully reduce cycle counts while preserving a high percentage of valid egonets, underscoring their effectiveness. Sample graphs generated for this scenario are presented in Figure 3 (Appendix E.3.2).

## 4.2 Fair Graph Generation

We evaluate GGDiff's performance on generating fair graphs. To quantify structural fairness, we adopt the notion of Dyadic Parity (DP) from Navarro et al. (2024), which posits that the probability of an edge existing should be independent of the sensitive attributes of the connected nodes. We report two metrics measuring the deviation from this ideal: $\Delta$DP, which captures global bias by aggregating the squared differences between average intra-group and inter-group connection strengths, and $\Delta$DP$_{node}$, a stricter, local metric that assesses node-level deviations to ensure that individual nodes maintain balanced connectivity across all sensitive groups.

To demonstrate the generality of our framework, we apply it to two distinct backbones: the GDSS graph diffusion model Jo et al. (2022) and the recent state-of-the-art GruM model Jo et al. (2024). For these experiments, we randomly assign sensitive attributes to the nodes of graphs generated from the QM9 dataset (see Appendix E for a more complex setup with community-based attributes). We compare our approach against the unconstrained baselines and the guidance framework proposed for DiGress (Vignac et al., 2023).

The results in Table 3 demonstrate GGDiff's superior performance and versatility. On the GDSS backbone, GGDiff-G attains an exceptionally low $\Delta$DP (0.0057), reducing the metric by $\sim 8.3\times$ versus the baseline. On the GruM backbone, GGDiff-C achieves the best overall fairness score among all configurations ($\Delta$DP of 0.0012), drastically improving upon the unconstrained GruM baseline (0.0516). In contrast, the guided DiGress baseline offers only marginal gains. These results confirm that GGDiff is not limited to a specific architecture but is a general framework capable of leveraging diverse backbones for fair graph generation.

Table 3: Fair graph generation on the QM9 dataset.

Table 4: Incomplete graph generation on the QM9 dataset with edge minimization.

| Method | $\Delta$DP | $\Delta$DP$_{node}$ |
| --- | --- | --- |
| GGDiff-G (GDSS) | 0.0057 | 0.0832 |
| GGDiff-C (GDSS) | 0.0427 | 0.1153 |
| GGDiff-Z (GDSS) | 0.0223 | **0.0519** |
| GGDiff-G (GruM) | 0.0090 | 0.1848 |
| GGDiff-C (GruM) | **0.0012** | 0.1177 |
| Uncons. (GGDS) | 0.0474 | 0.1206 |
| Uncons. (GruM) | 0.0516 | 0.1414 |
| DiGress Guidance | 0.0485 | 0.1445 |
| Uncons. (DiGress) | 0.0696 | 0.1560 |

| Method | % Unique | Num. Edges |
| --- | --- | --- |
| GGDiff-G (GDSS) | 62.62 | 9.4 ± 2.1 |
| GGDiff-C (GDSS) | 94.64 | 10.1 ± 2.0 |
| GGDiff-Z (GDSS) | 49.32 | 7.4 ± 2.9 |
| GGDiff-G (GruM) | 89.40 | 6.5 ± 2.3 |
| GGDiff-C (GruM) | 99.90 | 9.2 ± 1.1 |
| Uncons. (GDSS) | 96.39 | 10.8 ± 1.6 |
| Uncons. (GruM) | 99.95 | 9.2 ± 1.2 |
| DiGress Guidance | 94.42 | 10.6 ± 1.4 |
| Uncons. (DiGress) | 94.32 | 10.9 ± 1.4 |

### 4.3 INCOMPLETE GRAPH GENERATION

For the incomplete graph generation task, we compare our method, GGDiff, applied to both GDSS and GruM backbones, with DiGress (Vignac et al., 2023) on the QM9 dataset. DiGress enforces observed subgraphs using a masking procedure, as detailed in Vignac et al. (2023, Appendix E). This involves updating the adjacency matrix at each diffusion step as $\mathbf{A}_{t-1} = \mathbf{M} \circ \mathbf{A}_s + (\mathbf{1} - \mathbf{M}) \circ \mathbf{A}_t$, where $\mathbf{M}$ is a mask indicating observed entries and $\mathbf{A}_s$ represents the observed subgraph. We enforce this same masking within our method, and, to compare both methods beyond merely imposing an observed graph, we assess their ability to guide the generation towards graphs with as few edges as possible.

The results, presented in Table 4, demonstrate the advantage and flexibility of our framework. While DiGress barely reduces the number of edges (0.3 on average), GGDiff-G combined with GruM generates graphs with the lowest average number of edges (6.5), significantly outperforming the baselines while maintaining high sample uniqueness (89.4%). GGDiff-Z on GDSS also effectively minimizes edges but exhibits a stronger trade-off in sample diversity; we refer the reader to Appendix E.4 for a detailed analysis of this relationship between guidance strength and diversity.

Furthermore, we observe that performance is sensitive to the guidance strength $\lambda$; we provide a detailed analysis of sampling stability, convergence, and the trade-off between constraint satisfaction and chemical validity in terms of Fréchet ChemNet Distance (FCD) in Appendix E.4.

## 5 CONCLUSIONS

In this paper, we introduced GGDiff, a flexible framework for conditional graph generation, grounded in SOC. By casting guidance as a control problem, GGDiff enables plug-and-play conditioning of pre-trained diffusion models under both differentiable and black-box constraints. GGDiff unifies a range of existing guidance approaches, including gradient-based guidance and non-differentiable cases, under a single SOC-based formulation. Our method supports both hard and soft constraints without requiring gradient access or projection operators, making it broadly applicable across domains. Extensive experiments on structural, fairness, and topology-based constraints across different diffusion backbones demonstrate GGDiff's effectiveness and generality, outperforming prior work in handling complex, non-differentiable objectives. Despite its flexibility, GGDiff has limitations, as ZO optimization adds computational overhead and may suffer from high variance in challenging settings. Future work includes integrating GGDiff with discrete diffusion models to better handle discrete constraints, and exploring more efficient or adaptive ZO gradient estimators to improve scalability and performance.

### REPRODUCIBILITY STATEMENT

The experimental setups and results are detailed in Section 4 of the main paper. Further specifics, including comprehensive dataset descriptions, additional experimental details, and ablation studies, are provided in Appendix E.3. Furthermore, and to facilitate full reproducibility, we include a

complete codebase as supplementary material. This supplementary package contains clearly organized configuration files (e.g., YAML files) that detail all hyperparameters used across our experiments, enabling straightforward replication of our reported findings.

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

## A   HJB EQUATION

In this section, we give more details on our SOC formulation. The optimal control is given by

$$\mathbf{U}^*(\mathbf{G}_t^{\mathcal{C}}, t) = -\frac{g(t)}{\lambda} \nabla_{\mathbf{G}_t^{\mathcal{C}}} V_t^* \left( \mathbf{G}_t^{\mathcal{C}} \right)$$

where $V_t^*(\mathbf{G}_t^{\mathcal{C}})$ is the optimal value function Pavon (1989). For our problem, the optimal value function at time $t$ is given by

$$V_t^*(\mathbf{G}_t^{\mathcal{C}}) = \mathbb{E}_{p_t^*} \left[ \int_t^0 \lambda \frac{\|\mathbf{U}^*(\mathbf{G}_s^{\mathcal{C}}, s)\|_2^2}{2} \mathrm{d}s - r(\mathbf{G}_0^{\mathcal{C}}) \,\Big|\, \mathbf{G}_t^{\mathcal{C}} \right] \tag{13}$$

where $p_t^*$ denotes the optimal *controlled* distribution at time $t$ given by $p_t^*(\mathbf{G}) \propto \exp\left( \frac{-V_t^*(\mathbf{G})}{\lambda} \right) p_t^{\mathrm{pre}}(\mathbf{G})$ and $p_t^{\mathrm{pre}}$ is the prior (uncontrolled) distribution[1]. The value function $V_t^*$ solves the stochastic Hamilton-Jacobi-Bellman (HJB) equation Evans (2022), given by

$$\partial_t V_t^*(\mathbf{G}_t^{\mathcal{C}}) = \tag{14}$$
$$+ \left( \nabla_{\mathbf{G}_t^{\mathcal{C}}} V_t^*(\mathbf{G}_t^{\mathcal{C}}) \right)^T \boldsymbol{\mu}(\mathbf{G}_t^{\mathcal{C}}, t) - \frac{g(t)^2}{2\lambda} \left\| \nabla_{\mathbf{G}_t^{\mathcal{C}}} V_t^*(\mathbf{G}_t^{\mathcal{C}}) \right\|_2^2 + \frac{1}{2} g(t)^2 \Delta_{\mathbf{G}_t^{\mathcal{C}}} V_t^*(\mathbf{G}_t^{\mathcal{C}}),$$

with boundary condition $V_0(\mathbf{G}_0^{\mathcal{C}}) = r(\mathbf{G}_0^{\mathcal{C}})$, and where $\boldsymbol{\mu}(\mathbf{G}_t^{\mathcal{C}}, t) = \frac{1}{2}\mathbf{G}_t^{\mathcal{C}} - g(t)^2 \nabla_{\mathbf{G}_t^{\mathcal{C}}} \log p(\mathbf{G}_t^{\mathcal{C}})$.

---

[1]We assume here that the terminal time is 0 and the time runs backwards (so $t < 0$).

This equation is a *non-linear* partial differential equation (PDE), and the solution to the non-linear HJB equation is nontrivial. However, by applying an exponential transformation $\phi_t(\mathbf{G}_t^{\mathcal{C}}) = e^{-V_t(\mathbf{G}_t^{\mathcal{C}})}$, we can obtain the *linear* HJB equation, given by

$$-\partial_t\phi(\mathbf{G}_t^{\mathcal{C}},t) = \left(\nabla_{\mathbf{G}_t^c}\phi(\mathbf{G}_t^{\mathcal{C}},t)\right)^T \boldsymbol{\mu}(\mathbf{G}_t^{\mathcal{C}},t) + \frac{1}{2}g(t)^2\Delta_{\mathbf{G}_t^c}\phi(\mathbf{G}_t^{\mathcal{C}},t) \tag{15}$$

In particular, the Feynman-Kac formula is obtained as the solution of the linearized HJB equation in (15) (see Øksendal (2003) for the proof), given by

$$\exp\left(\frac{V_t^*(\mathbf{G})}{\lambda}\right) = \mathbb{E}_{p^{\mathrm{pre}}}\left[\exp\left(\frac{-r(\mathbf{G}_0^{\mathcal{C}})}{\lambda}\right)\ \Big|\ \mathbf{G}_t^{\mathcal{C}} = \mathbf{G}\right]. \tag{16}$$

This leads to an expression for the optimal control in terms of the reward function as given by (4).

**Stochastic optimal control for zero-shot controlled generation.** Recent methods have proposed the use of SOC for controlled generation Uehara et al. (2025); Li et al. (2024). In the context of music generation Huang et al. (2024), the authors propose a method to generate samples when likelihoods are non-differentiable. In Rout et al. (2024), a linear quadratic control was proposed for style transfer in image generation. More recently, a non-linear control formulation was introduced in Pandey et al. (2025) for image inverse problems. However, as far as we are concerned, the application of SOC for graph generation has not been explored yet.

# B BACKGROUND ON ZEROTH-ORDER OPTIMIZATION

In Section 3.2.2, we leverage ZO optimization for defining a surrogate gradient of the reward function. We propose three estimators in particular, where each one has its own properties. In this section, we expand on them.

**Two-point gradient estimator.** The two-point gradient estimator in (10) is the first one that we introduced. This estimator has a mean-squared error given by

$$\mathbb{E}[\|\hat{\nabla}r(\mathbf{G}_0) - \nabla r((\mathbf{G}_0))\|_2^2] = O(d)\|\nabla r(\mathbf{G}_0)\|_2^2 + O\left(\frac{\mu^2 d^3 + \mu^2 d}{\phi(d)}\right) \tag{17}$$

The proof can be found in Liu et al. (2018). The error in (17) sheds light on the behavior of this estimator. First, the second term depends on the parameter $\mu$: when this parameter gets smaller, the gradient estimate gets better. However, if $\mu$ becomes too small, then the effect of the guidance diminishes. Second, the first term depends on the dimension $d$. This imposes a variance which cannot be 0 even for small values of $\mu$.

**Multi-point gradient estimator.** The third estimator is based on the multi-point gradient estimate, which computes an average over random directions. This estimator has a mean-squared error given by

$$\mathbb{E}[\|\hat{\nabla}r(\mathbf{G}_0) - \nabla r((\mathbf{G}_0))\|_2^2] = O\left(\frac{d}{N}\right)\|\nabla f(\mathbf{x})\|_2^2 + O\left(\frac{\mu^2 d^3}{\phi(d)N}\right) + O\left(\frac{\mu^2 d}{\phi(d)}\right) \tag{18}$$

Compared to the two-point case, the error in (18) depends on the number of samples that are used to compute the average. In particular, the first two terms go to 0 when $N \to \infty$; the third term is independent of $N$, and corresponds to the approximation error between the true gradient and the smoothed version. However, it is controlled by the smoothing parameter $\mu$.

A summary of each estimator is shown in Table 5.

## B.1 UNBIASEDNESS OF THE ONE-POINT ESTIMATOR

For a Gaussian $p(\mathbf{U}_t)$, we have the following smoothed version of

$$r_\mu(.) = \mathbb{E}_{\mathbf{U}_t}[r(\mathbf{G}_t + \mu\mathbf{U}_t)] = \int r(\mathbf{G}_t + \mu\mathbf{U}_t)\mathcal{N}(\mathbf{U}_t; 0, \mathbf{I})\mathrm{d}\mathbf{U}_t.$$

Table 5: Comparison of ZO estimators for control direction optimization.

| Method | Variance | Reward evaluation |
|---|---|---|
| 2-Point Estimator | High | 2 |
| Best-of-$N$ Direction | Low | $N$ |
| Averaged Random Search | Moderate | $N+1$ |

By a change of variable $\mathbf{Y}_t = \mathbf{G}_t + \mu\mathbf{U}_t$, we get $\int r(\mathbf{Y}_t)\mathcal{N}(\mathbf{Y}_t; \mathbf{G}_t, \mu^2\mathbf{I})\frac{1}{\mu}\mathrm{d}\mathbf{Y}_t$. Then, the gradient w.r.t. $\mathbf{G}_t$ boils down to

$$\int r(\mathbf{Y}_t)\frac{\mathbf{Y}_t - \mathbf{G}_t}{\mu^2}\mathcal{N}(\mathbf{Y}_t; \mathbf{G}_t, \mu^2\mathbf{I})\mathrm{d}\mathbf{Y}_t = \int r(\mathbf{G}_t + \mu\mathbf{U}_t)\frac{\mathbf{U}_t}{\mu}\mathcal{N}(\mathbf{U}_t; 0, \mathbf{I})\mathrm{d}\mathbf{U}_t$$
$$= \frac{1}{\mu}\mathbb{E}_{\mathbf{U}_t}[r(\mathbf{G}_t + \mu\mathbf{U}_t)\mathbf{U}_t].$$

On the other side, the one-point estimate is $\frac{1}{\mu}r(\mathbf{G}_t + \mu\mathbf{U}_t)\mathbf{U}_i$. Hence, by taking expectation w.r.t. $\mathbf{U}_t$, we get the unbiased result.

## C    DIFFUSION MODELS FOR GRAPH GENERATION.

Graph diffusion models have been developed in both *continuous* and *discrete* domains.

**Continuous domain.**    The continuous formulation was introduced in EDP-GNN (Niu et al., 2020) to diffuse the graph topology, later extended in GDSS (Jo et al., 2022) to include node features, and further explored in the spectral domain (Luo et al., 2023; Minello et al., 2025). Lastly, in Zhou et al. (2024), the authors proposed to use a latent diffusion model. The main difference between the latent and the node-level formulation is that the former defines the diffusion process in a latent space. This requires an encoder-decoder pair to map from the node to the latent space and vice versa. Formally, the encoder $\mathbf{Z}_0 = \mathcal{E}(\mathbf{A})$ maps the adjacency matrix to a latent variable $\mathbf{Z}_0$. Then, $\mathbf{Z}_0$ follows a diffusion process similar to the one in Section 2.1, and then a decoder is used to generate an adjacency matrix $\hat{\mathbf{A}} = \mathcal{D}(\mathbf{Z}_0)$. At a high level, continuous models focus on capturing *global structure*

**Discrete domain.**    Discrete diffusion was introduced in DiGress (Vignac et al., 2023) by adapting the structured diffusion framework (Austin et al., 2021), framing generation as edge-wise classification to mitigate combinatorial complexity.

In a nutshell, each node and edge in the graph $\mathcal{G}$ is assumed to take on values from a fixed set of possible categories, and the model learns a categorical distribution for these variables. In other words, the graph represents a multivariate random variable, where each component follows a categorical distribution. Then, the forward process perturbs the graph by sampling from a modified, discrete, distribution, until it converges to a stationary state (such as a mask or uniform; see Austin et al. (2021)). The "reverse process", on the other hand, involves recovering the original graph by denoising this uniformly random structure step by step.

While discrete methods are well-suited for sparse graphs, they rely on mean-field approximations and lack gradients, limiting constrained generation. To address inference speed, EDGE (Chen et al., 2023) proposed using the empty graph as the stationary distribution, achieving faster sampling than DiGress but retaining the gradient limitation. Given these trade-offs, we focus on the continuous setting (see Appendix D for details).

## D    DIFFUSION MODELS FOR CONTROLLABLE GENERATION ON GRAPHS

Diffusion models have shown impressive results in unconstrained graph generation, either when considering discrete diffusion Vignac et al. (2023); Chen et al. (2023) or score-based approaches, i.e.,

considering a continuous relaxation (Niu et al., 2020; Jo et al., 2022). However, only a few works address the generation of graphs under constraints using diffusion models as priors.

**Guidance with differentiable rewards.** LGD Zhou et al. (2024) considers a classifier-free guidance approach Ho & Salimans (2021) in a latent space. To incorporate more complex constraints, in Madeira et al. (2024) the authors propose to use a projection operator combined with an edge-absorbing model. While effective, the projection operator is computationally demanding. PRODIGY Sharma et al. (2024), on the other hand, leverages continuous diffusion models and incorporates a projection operator. While the projection operator can be implemented efficiently using the bisection method, this only holds for simple constraints with a closed-form projection operator. This reliance limits its applicability since such operators are often unavailable. As a result, more complex constraints require computationally intensive solvers, significantly increasing runtime and restricting the scalability of PRODIGY to more general settings.

**Classifier-based Guidance for Graphs (CBG) Dhariwal & Nichol (2021).** This type of guidance employs an auxiliary network to approximate the guidance condition. To be more specific, the condition $\mathcal{C}$ is incorporated using a classifier/regressor $p_\theta(\mathcal{C} \mid \mathbf{G}_t)$ trained to predict the condition from a partially denoised graph $\mathbf{G}_t$. Then, the conditional score is as follow

$$\nabla_{\mathbf{G}_{t-1}} \log p^\gamma(\mathbf{G}_s \mid \mathcal{C}, \mathbf{G}_t) = \gamma \nabla_{\mathbf{G}_{t-1}} \log p_\theta(\mathcal{C} \mid \mathbf{G}_{t-1}) + \nabla_{\mathbf{G}_{t-1}} \log p_\theta(\mathbf{G}_{t-1} \mid \mathbf{G}_t), \quad (19)$$

where $\gamma > 0$ is a temperature parameter. This has been used in Vignac et al. (2023); **?**. The advantange of CBG compared to the guidance with a differentiable reward is that the model can handle non-differentiable ones (even those that are not defined). However, it requires a dataset with conditioning labels/target properties ot train the regressor. It is important to remark that, in many important applications, such datasets are unavailable. For example: in fairness applications, there is no dataset of "fair graphs" to train on; in molecular design, obtaining labeled datasets for novel properties can be prohibitively expensive. For many graph properties of interest, ground-truth labels simply do not exist at training time. Furthermore, it requires a regressor per task. In contrast, our method:

1. Leverages a single unconditional graph generation model that works in a plug-and-play fashion across different downstream tasks;

2. Only requires a reward/likelihood function that can evaluate generated samples (which need not be differentiable);

3. It does not require any labeled training data or model retraining for new conditioning tasks

**Classifier-free Guidance for Graphs.** This last case termed classifier-free guidance Ho & Salimans (2021) avoids training a separate classifier. Instead, it trains a single model to predict the reverse step *both conditionally* on $\mathcal{C}$ and *unconditionally* by randomly dropping the condition $\mathcal{C}$ during training.

Applying Bayes' rule to $p(\mathcal{C} \mid \mathbf{G}_{t-1})$ and differentiating gives the general relationship:

$$\nabla_{\mathbf{G}_{t-1}} \log p(\mathcal{C} \mid \mathbf{G}_{t-1}) = \nabla_{\mathbf{G}_{t-1}} \log p(\mathbf{G}_{t-1} \mid \mathcal{C}) - \nabla_{\mathbf{G}_{t-1}} \log p(\mathbf{G}_{t-1}).$$

Plugging the conditional $\nabla_{\mathbf{G}_{t-1}} \log p_\theta(\mathbf{G}_{t-1} \mid \mathcal{C}, \mathbf{G}_t)$ and unconditional $\nabla_{\mathbf{G}_{t-1}} \log p_\theta(\mathbf{G}_{t-1} \mid \mathbf{G}_t)$ model scores into Equation 19 and rearranging yields the classifier-free guidance formulation:

$$\nabla_{\mathbf{G}_{t-1}} \log p^\gamma(\mathbf{G}_{t-1} \mid \mathcal{C}, \mathbf{G}_t) = \gamma \nabla_{\mathbf{G}_{t-1}} \log p_\theta(\mathbf{G}_{t-1} \mid \mathcal{C}, \mathbf{G}_t) + (1-\gamma) \nabla_{\mathbf{G}_{t-1}} \log p_\theta(\mathbf{G}_{t-1} \mid \mathbf{G}_t). \quad (20)$$

Here, both terms belong to the same model, where the difference lies in using the conditioning variable ($p_\theta(\mathbf{G}_{t-1} \mid \mathcal{C}, \mathbf{G}_t)$) or not ($p_\theta(\mathbf{G}_{t-1} \mid \mathbf{G}_t)$). Notice that the advantage of CFG compared to CBG is that uses a single model for everything. However, this yields a bottleneck: the model is trained for a specific task. In particular, CFG requires training a separate model for each conditioning task, or access to a large multi-task dataset covering all properties of interest. This significantly limits flexibility and increases computational costs.

# E EXPERIMENTAL DETAILS

This appendix provides detailed information regarding the experimental setup used in this paper, including specifics about the datasets, computational resources utilized, and a comprehensive description of additional experiments conducted. The appendix is structured as follows: Section E.1

includes a description of the datasets used for evaluating GGDiff's performance. Section E.2, details the computational resources of the server where the experiments were run. Section E.4 presents the results of several ablation studies, both on the main hyperparameters of our method and on computational times. Section E.3 presents additional experimental results, with subsections dedicated to further details on constrained graph generation (Section E.3.1), generating star graphs (Section E.3.2), samples for the cycle minimization setup (Section E.3.3), fair graph generation (Section E.3.4) and link prediction (Section E.3.5).

## E.1 Datasets

We evaluate our proposed GGDiff framework and baselines on a selection of benchmark graph datasets, encompassing both generic network structures and molecular graphs. The datasets used in our experiments are described below:

1. **Ego-small**: This dataset comprises 200 small ego graphs extracted from the larger Citeseer network.

2. **Community-small**: Consisting of 100 synthetic graphs, this dataset features structures exhibiting distinct community partitions.

3. **Enzymes**: We use the protein graphs from the BRENDA enzyme database, totaling 587 graphs.

4. **QM9**: A molecular dataset containing approximately 133,000 small molecules. These molecules are composed of up to 9 heavy atoms, including Carbon (C), Nitrogen (N), Oxygen (O), and Fluorine (F).

5. **ZINC250k**: This large molecular dataset includes 250,000 drug-like molecules. The graphs represent molecules with 6 to 38 heavy atoms, incorporating Carbon (C), Nitrogen (N), Oxygen (O), Fluorine (F), Phosphorus (P), Chlorine (Cl), Bromine (Br), and Iodine (I).

## E.2 Computational resources

All experiments were conducted on a server equipped with an AMD EPYC 9634 84-Core Processor and 512GB of total physical memory (RAM). For accelerated computation, the server uses an NVIDIA GeForce RTX 4090 graphics processing units (GPUs), each featuring 24GB of dedicated video memory. The software environment runs on Ubuntu 24.04 LTS, with NVIDIA driver version 560.35.03 and CUDA version 12.6.

## E.3 Additional experiments

### E.3.1 Constrained Graph Generation

In this section, we provide additional details regarding the constrained graph generation experiments summarized in the main paper (see Table 1). For comparison purposes with prior work, we specifically focus on evaluating GGDiff's performance on the task of guiding the generated graphs towards fulfilling the constraints previously defined and utilized in Sharma et al. (2024). These constraints, designed to enforce specific structural properties, are presented in Table 6, along with their descriptions and mathematical formulations.

Table 6: Summary of Constraints from Sharma et al. (2024)

| Constraint Type | Limiting factor | Mathematical Formulation |
|---|---|---|
| Edge Count | Number of edges $|\mathcal{E}|$ | $|\mathcal{E}| = \mathbf{1}^\top \mathbf{A} \mathbf{1} \leq B$ for a given constant $B \geq 0$ |
| Triangle Count | Number of triangles | $\mathrm{tr}(\mathbf{A}^3) \leq T$ for a given constant $T \geq 0$ |
| Degree | Maximum Degree | $\max_i [\mathbf{A}\mathbf{1}]_i \leq D$ for a given constant $D \geq 0$ |

The values for constants $B$, $T$, and $D$ used for each dataset are selected based on those reported in Sharma et al. (2024) to ensure a direct comparison of method performance under identical constraint settings, and are given by those values fulfilled by 10% of the graphs in the test dataset.

Table 7: Metrics for the force stars constraint in the Ego small dataset.

| Method | % 1 Node | % Stars | % Stars & > 1 Node | % Valid Egonet | Edges over Star |
|--------|----------|---------|--------------------|----------------|-----------------|
| GGDiff-G | 0.78 | 53.12 | 52.34 | 96.09 | $1.08 \pm 2.61$ |
| GGDiff-L | 2.34 | 51.56 | 49.22 | 88.28 | $0.44 \pm 0.58$ |
| PRODIGY | 100.00 | 100.00 | 0.00 | 100.00 | $0.00 \pm 0.00$ |
| Uncons. | 0.78 | 24.22 | 23.44 | 99.22 | $1.86 \pm 2.64$ |

For each constraint, we empirically select the loss function from a pool of candidates, choosing the one that yields the best performance; the full set of options is provided in the code accompanying this submission. For differentiable guidance (Section 3.2.1), the choice is restricted to differentiable functions, typically involving $\ell_1$ or $\ell_2$ norms. For instance, an $\ell_2$ loss for the edge count constraint could be $(\mathbf{1}^\top \hat{\mathbf{A}}_0^{\mathcal{C}}(\mathbf{A}_t^{\mathcal{C}})\mathbf{1} - B)^2$. In contrast, the non-differentiable ZO guidance (Section 3.2.2) significantly expands the available loss functions. Examples include utilizing non-differentiable operations in the differentiable losses, like using the quantized adjacency via the entry-wise indicator function $\mathbb{1}(\hat{\mathbf{A}}_0^{\mathcal{C}}(\mathbf{A}_t^{\mathcal{C}}) > 0.5)$ in lieu of the estimate $\hat{\mathbf{A}}_0^{\mathcal{C}}(\mathbf{A}_t^{\mathcal{C}})$, or employing one-sided penalties such as $\max\{\mathbf{1}^\top \hat{\mathbf{A}}_0^{\mathcal{C}}(\mathbf{A}_t^{\mathcal{C}})\mathbf{1} - B, 0\}$.

It is important to remark that the choice of GGDiff variant depends on the nature of the graph constraint. For smooth constraints, such as simply minimizing edges, efficient gradient-based methods are suitable. However, most graph constraints, such as keeping the edge count below a fixed threshold, are combinatorial and non-differentiable in nature. Hence, our method based on ZO optimization is better suited.

### E.3.2 GENERATING STAR GRAPHS

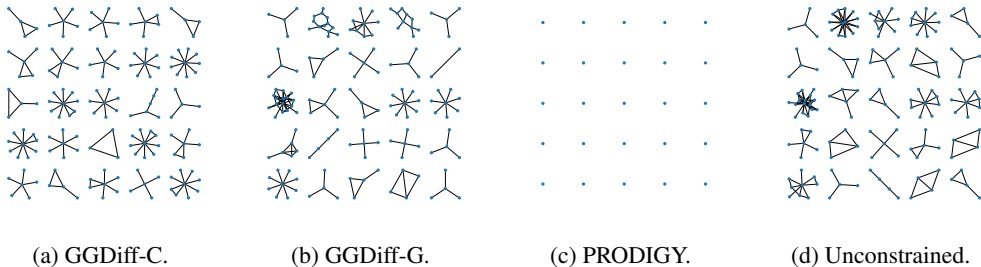

(a) GGDiff-C.  (b) GGDiff-G.  (c) PRODIGY.  (d) Unconstrained.

Figure 2: Samples for the force stars constraint in the Ego small dataset.

In this additional experiment, we investigate generating star graphs on the Ego small dataset, as another example of a complex constraint not covered within the PRODIGY framework. We enforced a zero-triangle constraint for PRODIGY as a proxy for the star structure. The results (Table 7) show that PRODIGY collapsed to generating only single-node graphs. In contrast, GGDiff doubled the output of star graphs over the unconstrained case and significantly reduced superfluous edges, all while maintaining high data fidelity, with 90% of its generated graphs being valid egonets (see also Figure 2).

### E.3.3 GENERATING GRAPHS MINIMIZING CYCLES

We show here the generated graphs for the experiment described in Section 4.1 for the cycle constraint.

(a) GGDiff-C.        (b) GGDiff-Z.        (c) Unconstrained.

Figure 3: Samples for the scenario minimizing cycles in the Ego small dataset.

### E.3.4 FAIR GRAPH GENERATION

In this appendix section, we provide further details regarding the fair graph generation experiments introduced in the main paper. These experiments evaluate GGDiff's ability to generate graphs that satisfy fairness criteria based on assigned sensitive attributes. To encourage fair graphs, we employ the same loss functions defined in Navarro et al. (2024). In this case, we highlight the potential of our approach to generate fair graphs in the community small graph dataset. Due to the lack of a pretrained model for DiGress on this dataset, and the complexity of the projection operator for PRODIGY, we report the comparison of our guidance framework against an unconstrained model. We investigate two distinct methods for assigning sensitive attributes to the nodes:

1. **Random assignment**: Sensitive attributes are assigned to nodes randomly.

2. **Community partitioning algorithm-based assignment**: Sensitive attributes are assigned to nodes based on the community structure identified by a community partitioning algorithm. This represents a more challenging scenario for generating fair graphs that are also valid Stochastic Block Models (SBMs). Since SBMs are characterized by a high density of intra-community edges and a low density of inter-community edges, aligning the sensitive attribute with community membership creates a direct tension: the fair loss function encourages the formation of edges between nodes with different attributes (i.e., nodes in different communities), while the underlying data distribution and the objective of generating valid SBMs favor the opposite.

The results in Table 8 demonstrate that our GGDiff methods effectively reduce the fairness metrics ($\Delta$DP and $\Delta$DP$_{node}$) and increase the number of edges between nodes with different sensitive attributes, indicating improved fairness. Crucially, these improvements are achieved while largely maintaining the generated graphs within the family of the prior distribution (SBMs), as reflected in the percentage of valid SBMs. Analyzing the results for the more complex community partitioning-based sensitive attribute assignment, our three GGDiff methods are still able to effectively reduce the fairness metrics compared to baselines, while largely maintaining a high percentage of valid SBMs. This quantitative improvement is visually corroborated by the sample graphs shown in Figure 4, where graphs generated by GGDiff show a higher density of edges connecting nodes of different sensitive attributes (indicated by node color) compared to the unconstrained case.

Table 8: Metrics for the fair graph generation with random sensitive attribute assignment and according to a community partition algorithm.

| Method | Random | | | Community | | |
|---|---|---|---|---|---|---|
| | $\Delta$**DP** | $\Delta$**DP**$_{node}$ | **% Valid SBM** | $\Delta$**DP** | $\Delta$**DP**$_{node}$ | **% Valid SBM** |
| GGDiff-G | $0.0026 \pm 0.0029$ | $0.0249 \pm 0.0125$ | 100.0000 | $0.3119 \pm 0.1289$ | $0.1892 \pm 0.0720$ | 77.3438 |
| GGDiff-C | $0.0035 \pm 0.0053$ | $0.0192 \pm 0.0121$ | 99.2188 | $0.3389 \pm 0.0763$ | $0.1931 \pm 0.0387$ | 98.4375 |
| GGDiff-Z | $0.0015 \pm 0.0020$ | $0.0061 \pm 0.0037$ | 95.3125 | $0.3451 \pm 0.0612$ | $0.1999 \pm 0.0498$ | 98.4375 |
| Uncons. | $0.0071 \pm 0.0145$ | $0.0295 \pm 0.0218$ | 99.2188 | $0.4133 \pm 0.0769$ | $0.2348 \pm 0.0449$ | 99.2188 |

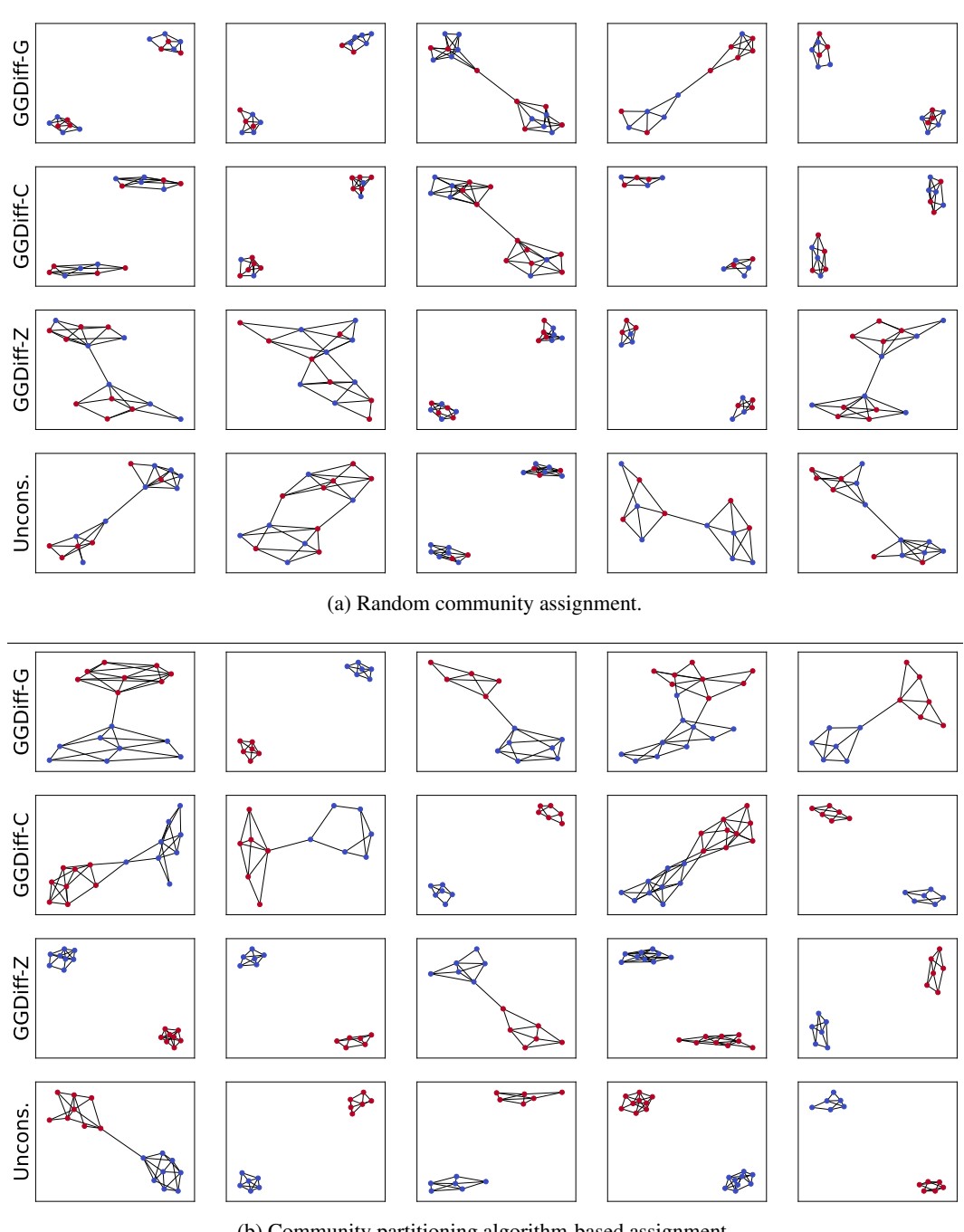

(a) Random community assignment.

(b) Community partitioning algorithm-based assignment.

Figure 4: Samples from the fair graph generation experiment. Colors represent node's sensitive attributes.

### E.3.5 INCOMPLETE GRAPH GENERATION

This section provides additional details on the link prediction experiments, also referred to as incomplete graph generation, in a different scenario than the one presented in the main paper. More precisely, we evaluate GGDiff's ability to generate graphs consistent with partially observed adjacency matrices: we assume that 50% of the entries of the adjacency matrix are observed and should be maintained in the generated graph. However, as opposed to the setting presented in the main paper, we relax the constraint, do not enforce the observed entries via masking, and we guide the generation towards respecting those entries via an $\ell_1$ or $\ell_2$ norm on the observed entries of the adjacency matrices. We conduct these experiments on two molecular datasets, QM9 and ZINC250k. We evaluate performance using two metrics: Accuracy, which measures the percentage of observed entries that are respected in the generated graphs; and % Unique, the percentage of generated molecules that are novel compared to the training set.

We investigate two scenarios for the observed entries: (i) observing a random 50% of all adjacency matrix entries (both existing edges and non-edges), and (ii) observing only a random subset of existing edges (entries equal to 1). This task is particularly relevant in domains like molecule generation, where there is often an interest in generating molecules that incorporate a specific predefined substructure (e.g., a benzene ring). Enforcing observed edges allows for the generation of molecules that respect such topological constraints.

The results for this scenario are presented in Table 9. For the observed entries case, the high accuracy values observed are significantly influenced by the correct generation of prevalent zero entries (non-edges) that were part of the observed subset. However, our GGDiff methods, particularly GGDiff-G and GGDiff-Z, demonstrate high accuracy in respecting the observed entries, both of them achieving almost 90% accuracy on QM9 and over 98% accuracy on ZINC250k.

We also observe a trade-off between accuracy and novelty in the QM9 dataset: as the accuracy in fixing observed entries increases, the percentage of novel molecules tends to decrease, suggesting that achieving very high fidelity to observed structure can lead to generating molecules highly similar to those in the test set. This tradeoff isn't observed in the ZINC250k dataset, likely due to the fact that the graphs are larger and therefore the model has more freedom to adapt to the observed entries.

For the second scenario, we observe only a random subset of existing edges in the adjacency matrix. Here, the accuracy metric specifically measures how well the generated graphs reproduce the observed edges. As expected, the accuracy values drop significantly compared to the first scenario because correctly generating existing edges is a more stringent condition than correctly generating non-edges in sparse graphs. However, the effect of GGDiff's guidance becomes strikingly apparent: our methods, particularly GGDiff-G and GGDiff-Z, achieve drastically increased accuracy in reproducing the observed edges on both the QM9 and ZINC250k datasets compared to the unconstrained baseline.

Sample graphs illustrating the results of the link prediction experiment are shown in Figures 5 and 6. In these visualizations, we use color and line style to indicate the status of observed entries in the generated graphs:

- **Solid green lines**: Observed edges that were successfully preserved in the generated graphs.

- **Solid red lines**: Observed edges that were *not* preserved in the generated graphs.

- **Dotted green lines**: Observed non-edge entries that were correctly preserved as non-edges.

- **Dotted red lines**: Observed non-edge entries that were *not* preserved (i.e., incorrectly generated as edges).

As observed in the figure, graphs generated by our GGDiff methods exhibit a clear prevalence of green lines (indicating high preservation of observed entries and edges), whereas the unconstrained case shows a greater number of red lines, highlighting its inability to reliably reproduce the specified topological constraints.

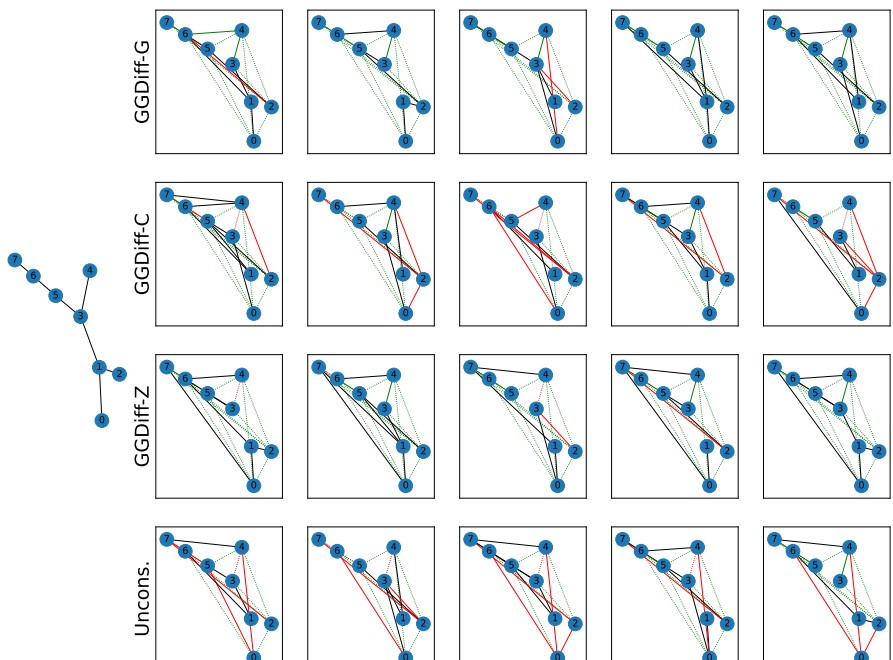

Figure 5: Samples generated for the incomplete graph generation experiment on QM9. The graph on the left is the test graph from which we observe the entries in its adjacency matrix. The generated graphs are represented in the rows, one for each of the methods. In the generated graphs, the solid green (red) lines are observed edges that were (not) preserved in the generated graphs, while dotted green (red) lines are observed entries not corresponding to an edge that were (not) preserved in the generated graphs.

Table 9: Metrics for the incomplete graph generation experiment.

| Method | Entries | | | | Edges | | | |
| | QM9 | | ZINC250k | | QM9 | | ZINC250k | |
| | Acc. (%) | % Unique | Acc. (%) | % Unique | Acc. (%) | % Unique | Acc. (%) | % Unique |
|---|---|---|---|---|---|---|---|---|
| GGDiff-G | 91.39 | 73.57 | 98.85 | 100.00 | 79.73 | 86.53 | 95.98 | 100.00 |
| GGDiff-C | 67.20 | 94.32 | 95.27 | 100.00 | 29.18 | 97.56 | 19.73 | 99.90 |
| GGDiff-Z | 88.72 | 90.88 | 98.44 | 100.00 | 41.66 | 92.71 | 85.59 | 100.00 |
| Uncons. | 61.51 | 97.86 | 93.43 | 100.00 | 23.40 | 97.81 | 8.45 | 100.00 |

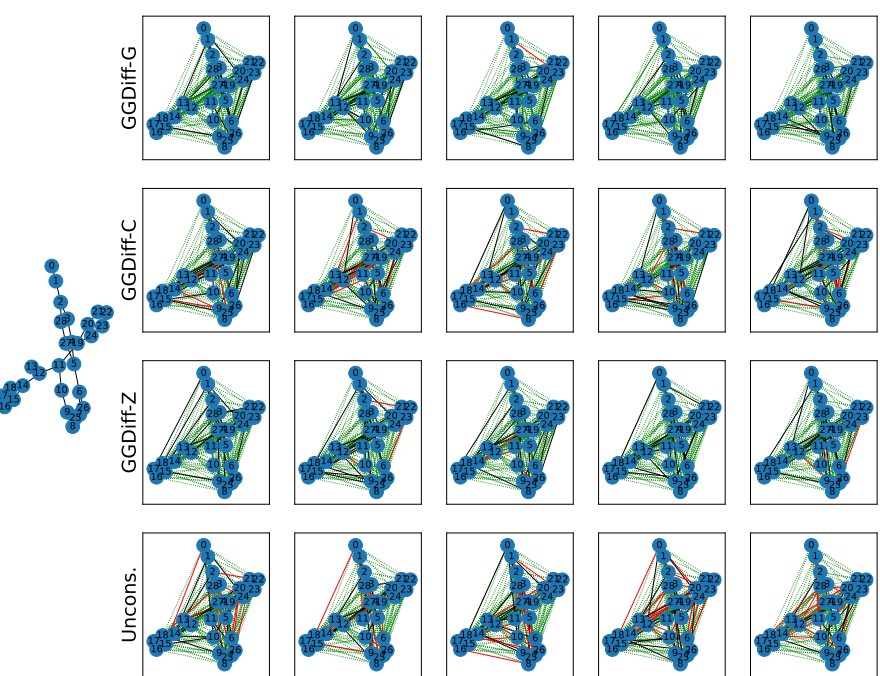

Figure 6: Samples generated for the incomplete graph generation experiment on ZINC250k. The graph on the left is the test graph from which we observe the entries in its adjacency matrix. The generated graphs are represented in the rows, one for each of the methods. In the generated graphs, the solid green (red) lines are observed edges that were (not) preserved in the generated graphs, while dotted green (red) lines are observed entries not corresponding to an edge that were (not) preserved in the generated graphs.

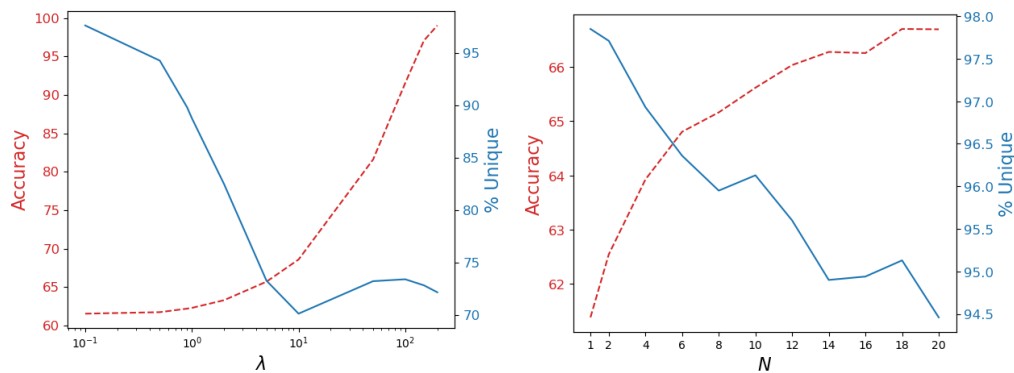

Figure 7: Ablation study on the hyperparameters $\lambda$ for GGDiff-G (left) and $N$ for GGDiff-C (right).

### E.4 ABLATION STUDIES

#### E.4.1 VARIATION WITH RESPECT TO $N$ AND $\lambda$

For the constrained-generation experiment, specifically the maximum-edge constraint, we report in Table 10 the ratio of graphs that satisfy it ($\mathrm{Val}_{\mathcal{C}} \in [0, 1]$ as defined in Sec. 4).

Table 10: Ablation study for different guidance strategies and hyperparameters on the maximum-edge constraint satisfaction ratio ($\mathrm{Val}_{\mathcal{C}}$).

| Parameter | Method | Parameter Value | | | | | | | | | | | | |
|---|---|---|---|---|---|---|---|---|---|---|---|---|---|---|
| | | **0** | **1** | **2** | **4** | **6** | **8** | **10** | **12** | **14** | **16** | **18** | **20** | |
| $N$ | | | | | | | | | | | | | | |
| | GGDiff-C | 0.156 | 0.156 | 0.445 | 0.562 | 0.602 | 0.633 | 0.656 | 0.602 | 0.656 | 0.625 | 0.594 | 0.609 | |
| | GGDiff-Z | 0.172 | 0.469 | 0.500 | 0.547 | 0.578 | 0.578 | 0.609 | 0.680 | 0.672 | 0.641 | 0.586 | 0.656 | |
| | | **0.0** | **0.01** | **0.1** | **0.5** | **1** | **5** | **10** | **50** | **100** | **150** | **200** | **250** | **300** |
| $\lambda$ | | | | | | | | | | | | | | |
| | GGDiff-G | 0.13 | 0.16 | 0.70 | 1.00 | 0.99 | 0.74 | 0.55 | 0.00 | 0.00 | 0.00 | 0.00 | 0.00 | 0.00 |

The table indicates that a larger $N$ strengthens the guidance, so more graphs satisfy the constraint under both Best-of-$N$ directions and the ZO-based method. For the hyperparameter $\lambda$ (step size of gradient-based approach) the values obtained for the $\mathrm{Val}_{\mathcal{C}} \in [0, 1]$ metric are as follows: as we raise $\lambda$ to 0.5, guidance strengthens and a higher proportion of graphs meet the constraint. Beyond that point, the algorithm becomes unstable and none of the graphs comply, underscoring the need for careful parameter tuning.

Figure 7 presents the result of the equivalent ablation study for the incomplete graph generation task, showing a clear trade-off between accuracy of the observed entries (this ablation follows the setup presented in Appendix E.3.5) and uniqueness. As the number of directions grows in the Best-of-$N$ setting, or the step size increases in the differentiable case, accuracy rises while the share of unique molecules falls. This is expected: larger $N$ or $\lambda$ amplify guidance in both methods. Notably, $\lambda$ has the stronger effect, boosting accuracy from about 60% (close to the unconstrained baseline) to nearly 100%, whereas the influence of $N$ is less pronounced.

We finalize this ablation study by also reporting the Fréchet ChemNet Distance (FCD), a domain-specific metric of chemical validity that offers a deeper view of our conditional generation quality. To probe the trade-off between constraint adherence and sample quality, we also ran an ablation on guidance strength $\lambda$, to highlight its impact on FCD.

The results, presented in Table 11, reveal that there is a "sweet spot" for guidance. As $\lambda$ increases, the FCD score first improves (decreases), reaching a minimum around $\lambda = 0.3$ before degrading. This demonstrates that overly aggressive guidance can pull the generated molecules too far from the learned data distribution, harming sample quality.

Table 11: Ablation study on the impact of guidance strength ($\lambda$) on Accuracy, Uniqueness, and Fréchet ChemNet Distance (FCD).

| Metric | Parameter Value | | | | | | | | | | | | | | | |
|---|---|---|---|---|---|---|---|---|---|---|---|---|---|---|---|---|
| | 0.001 | 0.01 | 0.05 | 0.1 | 0.2 | 0.3 | 0.4 | 0.5 | 0.6 | 0.7 | 0.8 | 0.9 | 1.0 | 10.0 | 50.0 | 100.0 |
| Accuracy (%) | 61.55 | 61.53 | 61.67 | 61.5 | 61.57 | 61.88 | 61.66 | 61.63 | 61.8 | 61.95 | 62.1 | 62.11 | 62.03 | 68.61 | 81.74 | 91.51 |
| % Unique | 97.68 | 97.7 | 97.77 | 97.74 | 97.45 | 97.04 | 95.89 | 94.58 | 93.06 | 91.78 | 90.99 | 89.58 | 88.79 | 69.93 | 73.23 | 73.16 |
| FCD | 2.55 | 2.59 | 2.56 | 2.51 | 2.44 | 2.37 | 2.30 | 2.50 | 2.70 | 2.96 | 3.12 | 3.49 | 3.76 | 8.62 | 9.39 | 9.07 |

### E.4.2 TIMING/COMPUTATIONAL COST

Finally, we conducted a new timing ablation study, where we measured the time required to sample new graphs for different values of the hyperparameter $N$. The primary source of computational overhead in GGDiff stems from the reward evaluation step, which is essential for our ZO guidance. Each guidance step requires one or more forward passes through the denoising model to generate a graph estimate on which the reward is computed. We quantify this conceptually in Table 5, which details the number of reward evaluations required for each guidance strategy. The cost of each individual evaluation is then determined by the underlying graph diffusion model and scales with the number of nodes and edges, which is why base sampling times increase when moving from smaller graphs like Ego-small to larger ones like Enzymes. To provide a concrete analysis of this overhead, we now present the timing ablation study. We measured the total sampling time while varying the number of candidate directions, $N$, for our Best-of-$N$ estimator.

Table 12: Timing ablation study measuring the total sampling time (in seconds) while varying the number of candidate directions, $N$, for our Best-of-$N$ (GGDiff-C) estimator.

| Dataset | Max. Node Number | Corrector | Sampling Time (s) for Number of Candidates ($N$) | | | | | |
|---|---|---|---|---|---|---|---|---|
| | | | **0** (No guide.) | **1** | **2** | **3** | **4** | **5** |
| Ego small | 18 | None | 18.37 | 39.42 | 59.25 | 84.97 | 110.62 | 119.39 |
| Enzymes | 125 | None | 43.88 | 87.83 | 130.71 | 173.33 | 217.17 | 260.16 |
| Community small | 20 | Lang (Steps 10) | 198.08 | 244.13 | 261.59 | 267.85 | 290.40 | 290.27 |
| QM9 | 9 | Lang (Steps 1) | 20.24 | 30.17 | 39.92 | 50.12 | 58.98 | 68.76 |
| ZINC250k | 38 | Lang (Steps 1) | 112.40 | 184.15 | 235.01 | 291.04 | 346.76 | 402.02 |

The results, presented in Table 12, offers two key insights. First, computational time increases linearly with $N$, as each candidate adds one reward evaluation. Second, and more importantly, our guidance's overhead is often marginal compared to other parts of the diffusion pipeline. For instance, on the Community small dataset, base sampling takes about $\sim$198 seconds, mostly due to 10 Langevin corrector steps. The total overhead for evaluating $N = 5$ candidates is under 100 seconds, a predictable cost that is small relative to the necessary, but expensive, corrector step. This shows GGDiff guidance is computationally light and usually not the primary bottleneck.

### E.4.3 STARTING GUIDANCE AT DIFFERENT POINTS IN THE DIFFUSION PROCESS

To evaluate the necessity of applying guidance throughout the diffusion process, we performed an ablation study varying the starting point of the guidance, denoted as $t_0$. We tested the GGDiff variants by applying guidance for the final $t_0$ steps of the trajectory. Table 13 summarizes the results in terms of $\Delta$DP and sampling time.

The results demonstrate that the duration of guidance is critical for both the performance and stability of GGDiff-G and GGDiff-Z. For GGDiff-G, applying guidance only for the final 100 steps yields a $\Delta$DP of $0.0401 \pm 0.0466$, offering negligible improvement over the unguided baseline. However, extending guidance to the full 1,000 steps significantly reduces $\Delta$DP to $0.0058$ and, crucially, reduces the standard deviation by approximately 84% (from $\pm 0.0466$ to $\pm 0.0076$), indicating much more consistent generation of fair graphs.

We note that this trade-off is method-dependent. GGDiff-C proves less sensitive to the guidance horizon, showing minimal improvement in fairness even when guided for the full trajectory, despite the linear increase in sampling time. Consequently, we adopted full-trajectory guidance ($t_0 = 1000$) as

the default for our main experiments to maximize fairness and reliability, accepting the computational cost associated with the iterative gradient calculation.

Table 13: Ablation study on the number of guided diffusion steps. We report the $\Delta$DP and Sampling Time (s) for varying guidance durations. Lower $\Delta$DP indicates better fairness.

| Method | Metric | 0 | 50 | 100 | 200 | 300 | $t_0$ 400 | 500 | 600 | 700 | 800 | 1000 |
|--------|--------|-----|-----|-----|-----|-----|-----|-----|-----|-----|-----|-----|
| GGDiff-G | $\Delta$ DP | 0.047 | 0.040 | 0.040 | 0.013 | 0.011 | 0.009 | 0.008 | 0.006 | 0.004 | 0.004 | 0.006 |
| | Time (s) | 81.96 | 89.17 | 96.64 | 111.27 | 125.39 | 139.85 | 154.14 | 168.91 | 182.84 | 198.00 | 226.40 |
| GGDiff-C | $\Delta$ DP | 0.047 | 0.047 | 0.048 | 0.047 | 0.047 | 0.046 | 0.047 | 0.047 | 0.046 | 0.047 | 0.043 |
| | Time (s) | 78.83 | 116.86 | 158.16 | 238.68 | 320.28 | 401.39 | 480.80 | 559.96 | 641.73 | 720.75 | 877.84 |
| GGDiff-Z | $\Delta$ DP | 0.048 | 0.036 | 0.036 | 0.036 | 0.036 | 0.035 | 0.029 | 0.019 | 0.018 | 0.014 | 0.022 |
| | Time (s) | 79.01 | 112.58 | 148.15 | 218.08 | 288.39 | 357.59 | 427.98 | 497.73 | 568.40 | 636.92 | 775.19 |

As observed in Table 13, the effectiveness of the guidance signal is strongly correlated with the number of guided steps. For GGDiff-G, applying guidance only during the final 10% of steps (100 steps) results in a $\Delta$DP of 0.0401, which is comparable to the unguided baseline (0 steps, $\Delta$DP 0.0473). However, extending the guidance to the full 1,000 steps results in a significant performance gain, reducing $\Delta$DP to 0.0058.

While reducing the number of guided steps does decrease sampling time linearly, the trade-off results in a substantial degradation of fairness properties. This suggests that the gradient guidance provided by GGDiff is necessary throughout the early and intermediate stages of the generation process to effectively steer the graph structure toward the desired fair distribution.

### E.4.4 GRADIENT ESTIMATION ACCURACY

To assess the variance and accuracy of our zeroth-order estimator, we compare the estimated gradient $\hat{\nabla} r(\mathbf{G}_t)$ against the true analytical gradient $\nabla r(\mathbf{G}_t)$. We utilize a differentiable reward function (from the fair graph generation task) to compute the ground truth gradient, while treating it as a black box for the ZO estimator.

We report the relative error, defined as $\|\frac{\nabla r}{\|\nabla r\|_2} - \frac{\hat{\nabla} r}{\|\hat{\nabla} r\|_2}\|_2$, as we vary the number of sampling directions $N$.

As illustrated in Figure 8, the relative error decreases as the number of directions $N$ increases. This demonstrates that the variance of the estimator can be mitigated by increasing the computational budget ($N$), ensuring that the guidance signal aligns sufficiently with the true direction of improvement.

## F RELATIONSHIP BETWEEN SOC AND RL

The SOC and RL formulations for diffusion models share the same underlying mathematical formulation: both can be viewed as solving a Markov Decision Process over the diffusion trajectory. In fact, our approach is closely related to the framework of *value-weighted sampling* described by Uehara et al. (2025, Section 6.2). In this appendix we would like to highlight key differences between this approach and ours, while acknowledging the connections between them.

When optimizing the policy (the transition kernel, and therefore, the score function), the two approaches differ fundamentally in their goal:

**RL-based approaches (e.g., DPOK (Fan et al., 2023), RLDF (Zhao et al., 2025))**

- **Goal**: Fine-tune/optimize the diffusion model parameters $\theta$ (the transition kernel) to maximize expected reward;
- **Requirements**: Access to training data and the ability to retrain the model;
- **Advantage**: Modify the model's distribution to better align with the reward;
- **Limitation**: Requires retraining for each new conditioning task; needs sufficient data for policy optimization.

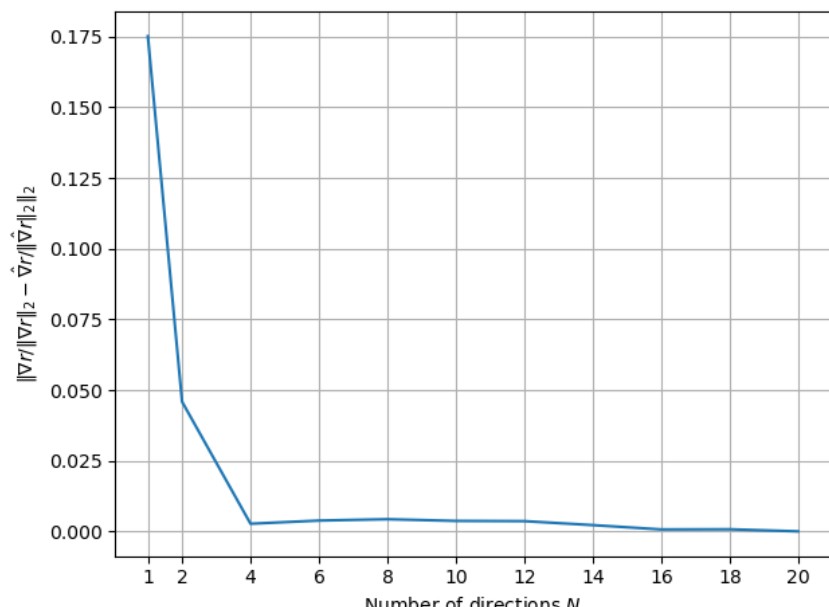

Figure 8: Evolution of the gradient estimation error $\|\nabla r(\mathbf{G}_t) - \hat{\nabla} r(\mathbf{G}_t)\|_2 / \|\nabla r(\mathbf{G}_t)\|_2$ with the number of directions $N$ on the fair graph generation experiment.

**SOC (our method)**

- **Goal**: Design an optimal controller $\mathbf{U}(\mathbf{G}_t^c, t)$ that guides a frozen pre-trained diffusion model

- **Requirements**: Only a pre-trained unconditional model and a reward/likelihood function

- **Advantage**: Plug-and-play at inference time; single model handles multiple tasks; no retraining needed

- **Limitation**: Constrained to modify the base process rather than fully optimizing it

### F.1 CONNECTION TO VALUE-WEIGHTED SAMPLING

Uehara et al. (2025) demonstrate guidance can be posed as a value-weighted sampling procedure, with value function given by

$$V^*(x_t, t) = \mathbb{E}_{x_0 \sim p(x_0 | x_t)}[\exp(r(x_0))]$$

and the optimal control is given by the gradient of $\log V^*(x_t, t)$. This is exactly the expression that we obtained, adapted to the discrete graph setting. Hence, value-weighted sampling with RL and SOC are equivalent. We took the SOC path because we believed is a more principled way of seeing this problem of controlling the trajectory of the diffusion model.

On the other hand, the reason why we use SOC rather than fine-tuning or policy optimization lies in the following facts:

1. **No task-specific training data**: In applications like fairness-aware graph generation or optimizing novel graph properties, we often lack the labeled datasets required for RL-based fine-tuning.

2. **Multi-task flexibility**: A single unconditional model can be steered toward different objectives at inference time by simply changing the reward function $r(\cdot)$, without any retraining.

3. **Inference-time control**: The SOC formulation provides an analytical solution (via the HJB equation and Feynman-Kac formula) for the optimal control [cf. (4)]. This reveals that

optimal guidance reduces to learning a value function, which can be done efficiently without access to training data.

4. **Practical efficiency**: For practitioners, our method means: train one unconditional model once, then apply it to arbitrary downstream tasks by specifying reward functions—no dataset collection or model retraining required.

In a nutshell, our SOC formulation is not a replacement for RL-based methods, but rather addresses a complementary and practically important setting: *conditional generation when task-specific training data is unavailable*. The key contribution is enabling a single pre-trained unconditional model to be steered toward arbitrary objectives at inference time, without retraining.

