# OpenReview forum: "Graph Guided Diffusion: Unified Guidance for Conditional Graph Generation"
_ICLR.cc/2026/Conference — Submitted to ICLR 2026_

### Official Review · Reviewer_cZPL · 2025-10-23

**Soundness:** 3
**Presentation:** 3
**Contribution:** 2
**Rating:** 4
**Confidence:** 4

**Summary:**

This paper proposes a novel framework named Graph Guided Diffusion (GGDiff) to address a key challenge in conditional graph generation with diffusion models—specifically, the difficulty of effectively guiding the generation process when the reward signal is arbitrary and non-differentiable (e.g., combinatorial rewards). The framework is plug-and-play, enabling zero-shot guidance of pre-trained diffusion models and supporting both differentiable and non-differentiable reward functions, achieving a good balance among computational efficiency, reward alignment, and sample quality.

**Strengths:**

1. Provides an alternative to reinforcement learning (RL) for generation with non-differentiable scoring functions.
2. Theoretically sound and clearly written.

**Weaknesses:**

1. Evaluation is limited, with comparisons against only a few baseline methods.
2. Generation efficiency under non-differentiable scoring functions is not assessed, which is critical for real-world applications.

**Questions:**

1. The most impactful application of conditional graph generation is molecular generation. Please include results on standard benchmarks such as QM9 and ZINC.
2. Please provide a comparison of sampling efficiency with and without GGDiff.
3. Is it necessary to apply guidance at every diffusion step? Please include an ablation study where GGDiff is applied only during the final 10% of steps.
4. Please evaluate the sample efficiency of GGDiff. Given that many reward functions are computationally expensive, it is important to analyze the trade-off between the number of reward function evaluations and generation quality.

---

> ### Author Response · Authors · 2025-11-21
> **Response to reviewer cZPL (1/2)**
>
> ## Molecular generation
>
> We fully agree with the reviewer that molecular generation is a highly impactful domain for conditional graph generation. We would like to clarify that our experimental evaluation extensively utilizes these standard benchmarks, though we apologize if their specific usage was not explicitly clear in every section of the original manuscript.
>
> Specifically, we have employed molecular datasets in the following experiments:
>
> - Fair graph generation (Section 4.2): These experiments were conducted using the QM9 dataset.
> - Incomplete graph generation (Section 4.3): this experiment was also performed using the QM9 dataset (although this was not indicated in the manuscript and we apologize for this lack of clarity. We have revised the manuscript to clarify that this experiment was performed on QM9).
> - Additional experiments for the incomplete graph generation (Appendix E.3.5): We evaluate our method on both QM9 and ZINC250K. Visual samples from these experiments are provided in Figures 5 and 6.
> - Ablation Studies: Several ablation studies (detailed in Section E.4) also utilize the QM9 dataset to benchmark performance.
>
> We appreciate the reviewer’s emphasis on this application. We believe our proposed guidance method is particularly valuable for molecular generation, especially in the context of incomplete graph generation (molecular inpainting). Our approach allows for the generation of valid molecules while strictly preserving a specific substructure of interest, a critical capability for drug discovery tasks.
>
> ## Comparison of sampling efficiency with and without GGDiff
>
> We thank the reviewer for this request. To address this, we updated the ablation study in Appendix E.4.1 to explicitly include the unguided baseline ($N=0$ and $\lambda=0$).
>
> Table 10 demonstrates the critical role of GGDiff in sampling efficiency:
> - Without guidance: The baseline model has a low success rate, with a constraint satisfaction ratio (Val$_{\mathcal{C}}$) ranging from 0.13 to 0.17. This implies that approximately 85% of generated samples fail to meet the criteria.
> - With GGDiff: activating guidance dramatically improves this yield. For GGDiff-G, the ratio improves from 0.13 to 1.00 (at $\lambda=0.5$), effectively ensuring that every sampled graph is valid. Similarly, GGDiff-C and GGDiff-Z show improvements from $\sim$ 0.16 to $\sim$ 0.68.
>
> These results confirm that GGDiff is essential for efficient sampling in constrained environments, transforming a generative process with low yield into one with high reliability.
>
> ## Evaluate the sample efficiency of GGDiff
>
> We thank the reviewer for raising this important point regarding the computational cost of guidance. We recognize that reward function evaluations can be expensive, and thus, finding the optimal balance between computational overhead and generation quality is crucial.
>
> To address this, we utilized the ablation study on the starting point of guidance ($t_0$) to analyze this trade-off. In our framework, the number of reward evaluations is directly proportional to $t_0$, and we can see this in the linear increase of computational time when increasing $t_0$.
>
> As detailed in Table 13 in Appendix E.4.3, which is the table presented in the response to the previous comment, we observe the following trade-offs:
>
> - Computational Cost: The sampling time increases linearly with the number of guided steps. For GGDiff-G, increasing guidance from 0 steps (baseline) to 1000 steps (full guidance) increases the sampling time by a factor of approximately 2.7 (from 81.9s to 226.4s).
> - Generation Quality: While the cost increases linearly, the quality improvement is non-linear and substantial. Applying guidance for only short durations (e.g., $t_0=100$) saves time (96.6s) but fails to improve fairness ($\Delta$DP $\approx$ 0.040). However, investing in more reward evaluations ($t_0 \ge 600$) reduces the fairness metric by an order of magnitude ($\Delta$DP $< 0.01$).
>
> We conclude that while GGDiff introduces a computational overhead, this investment is highly efficient in terms of quality return. The "cost" of the additional reward evaluations is necessary to traverse the guidance landscape effectively and achieve the target distribution.
>
> Finally, we further analyze computational scalability in Appendix E.4.2, specifically regarding the number of candidate directions $N$. As shown in Table 12, while runtime scales linearly with $N$, the guidance overhead is often secondary to other pipeline components. This confirms that GGDiff is a predictable and relatively lightweight addition that does not act as the primary bottleneck in complex diffusion pipelines.

---

> ### Author Response · Authors · 2025-11-21
> **Response to reviewer cZPL (2/2)**
>
> ## Lack of further baselines
>
> Please, refer to response to reviewer KRg4, Section "Comparison with Standard Guidance Methods".
>
> ## Sampling efficiency under non-differentiable scoring functions
>
> We respectfully direct the reviewer to Section 4.1 (Constrained Graph Generation), where we explicitly evaluate our method on cycle minimization. This task relies on a non-differentiable scoring function (counting cycles). As shown in Table 2, our method achieves competitive performance in this setting, demonstrating its effectiveness for optimization tasks involving non-differentiable constraints.
>
> ## Apply guidance at every diffusion step
>
> We thank the reviewer for suggesting this ablation study. To address this, we evaluated the performance of our methods when initiating guidance at varying points ($t_0$) in the reverse diffusion trajectory, ranging from the final 50 steps to the full 1,000 steps.
>
> As shown in the table below, applying guidance only during the final 10% of steps (100 steps) is insufficient for achieving high fairness. For GGDiff-G, limiting guidance to the final 100 steps results in a $\Delta$DP of $0.0401$, which is statistically similar to the unguided baseline ($0.0473$). Furthermore, restricted guidance leads to high instability: the standard deviation at 100 steps ($\pm 0.0466$) is nearly an order of magnitude higher than at 1,000 steps ($\pm 0.0076$).
>
> We also observe that sensitivity to the guidance horizon varies by method. While GGDiff-C shows limited improvement regardless of the number of guided steps, GGDiff-G and GGDiff-Z benefit substantially from full-trajectory guidance. Although the sampling time increases with the number of guided steps, the full 1,000-step guidance is necessary to achieve both the lowest $\Delta$DP and the highest consistency (lowest variance).
>
>
> |Method|Metric|0|50|100|200|300|400|500|600|700|800|1000|
> |---|---|---|---|---|---|---|---|---|---|---|---|---|
> |GGDiff-G|$\Delta$DP|0.0473 $\pm$ 0.0537|0.0403$\pm$0.0466|0.0401$\pm$0.0466|0.0127$\pm$0.0171|0.0105$\pm$0.0145|0.0093$\pm$0.0123|0.0081$\pm$0.0113|0.0058$\pm$0.0080|0.0043$\pm$0.0058|0.0043$\pm$0.0050|0.0058$\pm$0.0076|
> |GGDiff-G|Sampling time|81.96|89.17|96.64|111.27|125.39|139.85|154.14|168.91|182.84|198.00|226.40|
> |GGDiff-C|$\Delta$DP|0.0472$\pm$0.0519|0.0472$\pm$0.0509|0.0475$\pm$0.0537|0.0470$\pm$0.0519|0.0468$\pm$0.0520|0.0463$\pm$0.0501|0.0468$\pm$0.0525|0.0466$\pm$0.0519|0.0461$\pm$0.0512|0.0470$\pm$0.0513|0.0427$\pm$0.0479|
> |GGDiff-C|Sampling time|78.83|116.86|158.16|238.68|320.28|401.39|480.80|559.96|641.73|720.75|877.84|
> |GGDiff-Z|$\Delta$DP|0.0479$\pm$0.0543|0.0355$\pm$0.0382|0.0363$\pm$0.0386|0.0362$\pm$0.0384|0.0357$\pm$0.0387|0.0349$\pm$0.0367|0.0288$\pm$0.0320|0.0185$\pm$0.0209|0.0184$\pm$0.0211|0.0144$\pm$0.0174|0.0223$\pm$0.0242|
> |GGDiff-Z|Sampling time|79.01|112.58|148.15|218.08|288.39|357.59|427.98|497.73|568.40|636.92|775.19|
>
> To demonstrate the generality of this finding, we conducted an equivalent ablation for the incomplete graph generation task (we created a new Appendix E.3.5 for experimental details). The results, reported in the table below, exhibit an even more pronounced dependency on full-trajectory guidance. For GGDiff-G, applying guidance for the final 700 steps yields negligible improvement over the unguided baseline (Accuracy remains $\approx$ 62-64%). However, extending guidance to the early stages of the process ($t_0 \ge 800$) triggers a sharp performance jump, reaching 97.03% accuracy with full guidance. This confirms that for structural constraints, the guidance signal is critical during the initial formation of the graph topology (early diffusion steps) and cannot be restricted to the final denoising phase, even at the cost of an increased sampling time.
>
> |Method|Metric|0|50|100|200|300|400|500|600|700|800|1000|
> |---|---|---|---|---|---|---|---|---|---|---|---|---|
> |GGDiff-G|Accuracy|61.61|61.78|62.29|62.2|62.28|62.55|62.87|62.77|63.97|95.82|97.03|
> |GGDiff-G|Sampling time|77.77|84.72|91.35|105.1|118.34|131.6|145.38|159.29|172.93|184.9|212.52|
> |GGDiff-C|Accuracy|61.56|61.54|61.66|61.53|61.53|61.54|61.59|61.74|61.5|61.45|66.39|
> |GGDiff-C|Sampling time|78.34|105.64|133.77|189.96|246.5|303.08|359.44|417.41|474.06|530.67|640.46|

---

### Official Review · Reviewer_KRg4 · 2025-10-24

**Soundness:** 2
**Presentation:** 2
**Contribution:** 2
**Rating:** 4
**Confidence:** 3

**Summary:**

This paper tackles the problem of conditional graph generation and proposes a unified framework grounded in stochastic optimal control (SOC). The key contribution lies in introducing a gradient-free approach that avoids the intractability of gradient-based optimization through the use of zero-order optimization techniques.

The proposed framework can handle both differentiable and non-differentiable reward functions. Within this framework, the authors present a family of methods: a standard gradient-based approach and several gradient estimators based on single-point, best-of-N, and multi-point (average random search) strategies. The methods are evaluated on multiple datasets and tasks.

**Strengths:**

Expect for the experimental section, the paper is clearly written and easy to follow, even for readers less familiar with stochastic optimal control. The presentation is structured and well-motivated.

The proposed method is conceptually sound and supported by rigorous derivations. The framework is general and theoretically appealing, providing a principled approach to conditional generation with non-differentiable objectives.

The empirical evaluation covers a wide range of datasets and tasks, illustrating the adaptability of the proposed approach.

**Weaknesses:**

### Comparison with Standard Guidance Methods
While the framework is motivated as an alternative to existing guidance-based conditional generation methods, the paper does not include direct comparisons with standard guidance approaches for graphs, such as classifier guidance or classifier-free guidance. Including these as baselines in the experimental section would substantially strengthen the empirical validation and support the paper’s claims.

In addition, the paper argues that these guidance methods require the differentiability of the target property. I respectfully disagree with this claim. Classifier-free guidance, in particular, does not inherently require differentiability of the targeted property. Even for classifier guidance, one can train a differentiable surrogate model to approximate a non-differentiable objective, which can serve as a practical workaround.
It would considerably reinforce the paper’s contribution to demonstrate empirically that such approaches fail or underperform under the same conditions, thereby highlighting the advantages of the proposed method more convincingly.

### Evaluation and Baselines
In many experiments, the proposed method is compared against only a single alternative model, whereas several other strong guidance methods exist in the literature. Expanding the comparison to include more recent or diverse methods would provide a clearer picture of the strengths and weaknesses of the proposed approach.

Moreover, Table 4 suggests that the method’s improved effectiveness may come at the cost of reduced diversity (as reflected by lower uniqueness). This trade-off deserves further analysis and discussion. It would also be helpful to specify which dataset was used for the experiments reported in Table 4, as this information is currently unclear.

### Model Backbone
The experimental setup builds upon GDSS, a relatively early diffusion-based model for graphs. As the field has advanced significantly, it would strengthen the empirical validation to evaluate the proposed approach on more recent and robust backbones, such as Grum or CatFlow [1]. Demonstrating that the proposed guidance mechanism is effective across different backbones would further support its generality.

### Methodological Limitations
Graphs are inherently discrete structures, and recent state-of-the-art models (e.g., DeFog [2], SID [3]) are based on discrete diffusion. However, the proposed framework relies on continuous diffusion backbones, which limits its applicability to a subset of graph generation problems. Exploring extensions or adaptations for discrete generative processes could significantly broaden the impact of this work.

Additionally, the proposed multi-point estimators appear computationally demanding, as indicated in Appendix E.

### Clarity and Presentation
While the paper is well-written overall, it can sometimes be difficult to distinguish which components are novel contributions and which are directly drawn from the stochastic optimal control literature. Providing clearer separation between prior work and original developments would enhance the paper’s readability.

Unlike the rest of the paper, the evaluation section is somewhat difficult to follow. Several pieces of information are missing (for example, the dataset used for Table 4), some elements are only indirectly referenced (such as the evaluation metrics described in Section 4.2), and certain notations (e.g., $\Delta$ MMD) are not clearly defined. Improving the clarity and completeness of this section would greatly enhance the overall readability and impact of the paper.

--------

[1] Floor Eijkelboom, Grigory Bartosh, Christian A. Naesseth, Max Welling, and Jan-Willem van de
Meent. Variational flow matching for graph generation. In The Thirty-eighth Annual Confer-
ence on Neural Information Processing Systems, 2024. URL https://openreview.net/
forum?id=UahrHR5HQh.

[2] Yiming Qin, Manuel Madeira, Dorina Thanou, and Pascal Frossard. Defog: Discrete flow matching
for graph generation. In Forty-second International Conference on Machine Learning, 2025. URL
https://openreview.net/forum?id=KPRIwWhqAZ.

[3] Y. Boget. Simple and critical iterative denoising: A recasting of discrete diffusion in graph generation. In Proceedings of the 42th International Conference on Machine Learning, Proceedings of Machine Learning Research. PMLR, July 2025.

**Questions:**

Please, see suggestion in weaknesses

---

> ### Author Response · Authors · 2025-11-21
> **Response to reviewer KRg4 (1/3)**
>
> ## Comparison with Standard Guidance Methods
>
> We apologize for the lack of clarity regarding the baseline implementation. In the experiments involving DiGress (Sections 4.2 and 4.3), we utilized classifier guidance with the ground-truth reward function. Since the reward functions in these experiments are fully differentiable, we computed the guidance gradients directly via backpropagation through the analytical reward, eliminating the need for a trained surrogate model.
>
> **Regarding Classifier-Free Guidance (CFG)**
> We agree with the reviewer that CFG does not require a differentiable property model. However, CFG has fundamental requirements that limit its applicability:
>
> 1. Labeled training data: CFG requires a dataset with conditioning labels/target properties. In many important applications, such datasets are unavailable. For example:
>
>    - In fairness applications, there is no dataset of "fair graphs" to train on.
>    - In molecular design, obtaining labeled datasets for novel properties can be prohibitively expensive.
>    - For many graph properties of interest, ground-truth labels simply do not exist at training time.
>
>
> 2. Task-specific training: CFG requires training a separate model for each conditioning task, or access to a large multi-task dataset covering all properties of interest. This significantly limits flexibility and increases computational costs.
>
> In contrast, our method:
>
> 1. Leverages a single unconditional graph generation model that works in a plug-and-play fashion across different downstream tasks
> 2. Only requires a reward/likelihood function that can evaluate generated samples (which need not be differentiable)
> 3. Does not require any labeled training data or model retraining for new conditioning tasks
>
> **Regarding Classifier Guidance with Surrogate Models**
> The reviewer correctly notes that one could train a differentiable surrogate model for non-differentiable objectives. However, this approach still requires labeled training data for the target property, and also it requires a new classifier for each downstream task.
>
> We acknowledge that direct empirical comparisons would strengthen the paper. However, we note that that classifier-free guidance requires fundamentally different training setups (conditional vs. unconditional models)
> For our experimental tasks (e.g., fairness, spectral properties), appropriate labeled training datasets do not exist for training CFG/classifier guidance baselines. Creating such datasets would require solving the very problem our method addresses
>
> We believe the key contribution of our work is enabling conditional generation without task-specific training data, which addresses a complementary set of problems to CFG. Similar to the image case, our method is **plug-and-play**, meaning that can be used to any downstream application without any additional fine-tuning or training. We will clarify this positioning in the revised manuscript and discuss the trade-offs between approaches more explicitly.
>
> ## Effectiveness vs Diversity trade-off
>
> We thank the reviewer for these insightful observations.
>
> 1. Dataset Clarification: We apologize for the omission. The experiments reported in Table 4 were conducted using the QM9 dataset. We have updated the caption of Table 4 and the corresponding main text (Section 4.3) to explicitly state this.
> 2. Diversity vs. Effectiveness Trade-off: We fully agree with the reviewer that there is an inherent trade-off between constraint satisfaction (effectiveness) and sample diversity (uniqueness). As the guidance signal becomes stronger, the generative process is more aggressively steered toward the mode of the conditional distribution, naturally reducing the variance of the outputs.
>
> Regarding the latter point, we explicitly analyze this phenomenon in Appendix E.4.1 and Figure 7. As detailed in our discussion:
> - Increasing the guidance strength ($\lambda$) or the number of evaluated directions ($N$) consistently improves the accuracy of the observed entries/constraints.
> - However, this improvement correlates with a decrease in the share of unique molecules.
> - For example, increasing $\lambda$ boosts accuracy from $\sim$ 60% (baseline) to nearly 100%, but causes a corresponding drop in uniqueness.
>
> We believe this behavior is expected in controllable generation: as the feasible solution space shrinks to satisfy strict constraints, the diversity of valid solutions naturally decreases. We have ensured this discussion is clearly referenced in the main text.

---

> ### Author Response · Authors · 2025-11-21
> **Response to reviewer KRg4 (2/3)**
>
> ## Model Backbone
>
> We thank the reviewer for this constructive suggestion. We agree that demonstrating the efficacy of our guidance mechanism on modern backbones is crucial for establishing its generality.
>
> To address this, we have extended our experimental evaluation to include GruM, a state-of-the-art diffusion model for graphs. We attempted to include CatFlow as well; however, we were unable to reproduce the necessary baselines as the official code and pre-trained checkpoints are not currently available.
>
> The results, now included in Tables 3 and 4 and reproduced below, strongly support the generality of our approach across different tasks:
>
> - Fair Graph Generation: GGDiff successfully steers the GruM backbone toward fairer distributions. Notably, GGDiff-C combined with GruM achieves the best overall performance among all tested configurations ($\Delta$DP of $0.0012$), significantly outperforming the unconstrained GruM baseline ($0.0516$).
>
> |Method|$\Delta$DP|$\Delta\text{DP}_{\text{node}}$|
> |---|---|---|
> |GGDiff-G (GDSS)|0.0057|0.0832|
> |GGDiff-C (GDSS)|0.0427|0.1153|
> |GGDiff-Z (GDSS)|0.0223|**0.0519**|
> |GGDiff-G (GruM)|0.0090|0.1848|
> |GGDiff-C (GruM)|**0.0012**|0.1177|
> |Uncons. (GGDS)|0.0474|0.1206|
> |Uncons. (GruM)|0.0516|0.1414|
> |DiGress Guidance|0.0485|0.1445|
> |Uncons. (DiGress)|0.0696|0.1560|
>
> - Incomplete Graph Generation: In the edge minimization task, GGDiff-C applied to GruM achieves a 99.9% uniqueness rate, surpassing the best GDSS-based variant (94.6%) while our guidance framework effectively minimizes the number of edges over the unconstrained baseline (6.5 vs 9.2 for GGDiff-G).
>
> |Method|% Unique|Num. Edges|
> |---|---|---|
> |GGDiff-G (GDSS)|62.62|9.4$\pm$2.1|
> |GGDiff-C (GDSS)|94.64|10.1$\pm$2.0|
> |GGDiff-Z (GDSS)|49.32|7.4$\pm$2.9|
> |GGDiff-G (GruM)|89.40|6.5$\pm$2.3|
> |GGDiff-C (GruM)|99.90|9.2$\pm$1.1|
> |Uncons. (GDSS)|96.39|10.8$\pm$1.6|
> |Uncons. (GruM)|99.95|9.2$\pm$1.2|
> |DiGress Guidance|94.42|10.6$\pm$1.4|
> |Uncons. (DiGress)|94.32|10.9$\pm$1.4|
>
>
> These findings validate the plug-and-play nature of our framework. Since GGDiff operates purely during sampling, it can be seamlessly applied to any pre-trained diffusion backbone without modification, leveraging the superior generative capabilities of modern architectures to achieve state-of-the-art performance on downstream tasks.
>
>
> ## Clarity and Presentation
>
> We thank the reviewer for these valuable suggestions regarding the presentation and clarity of the manuscript. We have revised the text to improve readability and completeness.
>
> 1. Distinguishing Novelty from Prior Work:
>    - The primary novelty of our work is the development of a plug-and-play algorithm for constrained graph generation that operates independently of the training data distribution and can incorporate general, non-differentiable constraints into any pre-trained, unconstrained model. Unlike prior works such as DiGress, which relies on classifier-based guidance, which requires data from the posterior/constrained distribution; or PRODIGY, which is limited to simple constraints that can be implemented as a projection, our method offers a universally applicable guidance mechanism.
>    - We achieve this by: (i) framing conditional generation as a Stochastic Optimal Control (SOC) problem, which provides a principled, rigorous foundation for defining guidance, and (ii) introducing a ZO optimization gradient method to efficiently handle the non-differentiable rewards associated with general constraints. In fact, combining these two tools to tackle graph constrained generation represents a novel approach.
> Crucially, ZO optimization unifies existing heuristic techniques (like best-of-N sampling) under a formal, gradient-free paradigm, enabling a structured approach to estimator selection and computational budget management. The connection we establish between diffusion guidance and the formal ZO literature represents a significant step towards developing truly general and flexible conditional generation mechanisms.
>
> 2. Evaluation Section Improvements: We have addressed the specific omissions and clarity issues in the experiments:
>    - Dataset Specification: We have clarified in the caption of Table 4 and the corresponding text that the Incomplete Graph Generation experiments were performed on the QM9 dataset.
>    - Metric Definitions (MMD): We have added a formal definition of the metrics used in Section 4.1. Specifically, we now define $\Delta$ MMD (adopted from PRODIGY) as the difference between the MMD of the unconstrained baseline and the constrained generated graphs, providing a clear measure of distributional shift.
>    - Fairness Metrics: In Section 4.2, we have expanded the description of the fairness metrics ($\Delta$DP and $\Delta$DP$_{\text{node}}$) to ensure the section is self-contained, rather than relying solely on external references.

---

> ### Author Response · Authors · 2025-11-21
> **Response to reviewer KRg4 (3/3)**
>
> ## Methodological Limitations
>
> Thanks for raising this point. Indeed, extending guidance techniques to the discrete setting can significantly broaden the impact of the work. However, this is a challenging open problem: in discrete diffusion models, each denoising distribution factorizes conditionally independently, meaning the model generates each edge independently at each step. To incorporate a guidance term $p(y \mid \mathbf{G}_t)$, this likelihood would need to factorize over edges, which generally does not hold.
>
> Nevertheless, we are actively working on a guidance mechanism based on **Mirror Langevin Dynamics** [1, 2], which is well suited to the discrete setting that samples in a logit space. Specifically, the sampling is performed in an unconstrained dual space (defined by a mirror map), while the actual variables live in a constrained primal space (the simplex in the discrete case for graphs).
> For the negative-entropy mirror map (the natural geometry for categorical distributions) the simplex is mapped to the unconstrained logit domain, which avoids explicit projections like in PRODIGY.
>
> The idea is as follows: the process is performed at each noise level $t$ after generating a few unconditional graphs $\mathbf{G}_t$:
>
>
> 1. **Initialization:** Relax the unconditional graph $\mathbf{G}_t \in \{0,1\}^{N \times N \times K}$ to the interior of the probability simplex.
>
> 2. **MLD Iteration:** Run $L$ steps of the MLD update, targeting the posterior distribution. This update performs a step in the dual logit space and projects back to the simplex:
>
> $$
> \\mathbf{G}\_{l+1} = \\nabla \\phi^\* \\left( \\nabla\\phi(\\mathbf{G}\_l) + \\gamma\_l \\nabla\_{G\_l} r(\\mathbf{G}\_l)\\, \\mathrm{d}t + \\sqrt{2\\gamma\_l \\nabla^2 \\phi(\\mathbf{G}\_l)}\\, \\mathrm{d}\\mathbf{Z}\_l \\right), \\qquad \\mathbf{Z}\_l \\sim \\mathcal{N}(0, I)
> $$
>
> Mirror Map Details:
>    - The **Mirror Map** is $\nabla \phi(\mathbf{G}) = \mathbf{1} + \log \mathbf{G}$.
>    - The **Inverse Mirror Map** (used via $\\nabla \\phi^\*$) is the **Softmax** function: $\\mathbf{G}\_{L} = \\text{Softmax}(\\mathbf{G}\_{L, \\text{dual}})$
>    - The **Hessian** is:
> $$
> \nabla^2\phi(\mathbf{G}) = \mathrm{diag}(1/\mathbf{G})
> $$
>    - The gradient of the reward $\nabla r(\mathbf{G}_l)$ is approximated using a **Zero-Order (ZO) gradient**.
>
>
> 3. **Final Discretization:** The resulting probability tensor is projected back to the discrete space via:
>     $$
>     G^\star_{ij} = \arg\max_{k} G_{ij,k}.
>     $$
>
>
> In summary, the idea can be thought as sampling version of the projection defiend in PRODIGY, but supporting more general rewards (non-differentiable) given that we can deploy our ZO gradient in the Langevin sampling.
>
> [1] Zhang, K. S., Peyré, G., Fadili, J., & Pereyra, M. (2020). Wasserstein control of mirror langevin monte carlo. In Conference on learning theory.
> [2] Hsieh, Y. P., Kavis, A., Rolland, P., & Cevher, V. Mirrored langevin dynamics. In NeurIPS 2018.

---

> > ### Comment · Reviewer_KRg4 · 2025-11-26
> >
> > Thank you to the authors for their responses.
> >
> > ## Comparison with Baselines
> >
> > I agree with the explanation provided. However, I believe that the potential advantages of your method do not remove the need for empirical comparison against existing approaches. These methods should still be included as baselines.
> >
> > In addition, it would strengthen the paper to include a more comprehensive discussion of the various guidance techniques, including their respective advantages and limitations.
> >
> > ## Evaluation
> >
> > Several issues remain in the evaluation:
> >
> > * Most metrics are reported without standard deviations, which makes it difficult to reliably interpret performance. In the few cases where standard deviations are provided (e.g., Table 4), differences appear to be small or not statistically significant.
> > * In Table 1, the reported ∆MMDs obtained using your guidance methods are worse than those of the unconditional models.
> > * The edge-minimization experiment remains unclear. As currently presented, why is the “no-edge” solution not a trivial optimum?
> > * No alternative guidance methods (other than Prodigy) using the same backbone are included for comparison, which limits the fairness and interpretability of the results.
> >
> > ## Guidance for Discrete Models
> >
> > I am also not yet convinced about how conditional independence in your formulation differs from the continuous case. Additional clarification on this point would be very helpful.

---

> > > ### Author Response · Authors · 2025-12-03
> > > **Response to reviewer KRg4 - Comparison with Baselines (1/3)**
> > >
> > > ### Comparison with Baselines
> > >
> > > We agree with the reviewer that an empirical comparison with existing baselines is essential to validate our method's effectiveness. To address this, we have replicated the conditional generation experimental setup described in **DiGress (Section 7.3)** on the QM9 dataset.
> > >
> > > In this setting, we condition the generation on two properties: the Dipole moment ($\mu$) and the Highest Occupied Molecular Orbital (HOMO). The results are presented in the table below.
> > >
> > > We highlight three key observations:
> > >
> > > 1. **Direct Guidance vs. Surrogate:** Unlike DiGress, which requires training an auxiliary regression model (a surrogate) to approximate gradients for these properties, **GGDiff-Z** computes the true property value directly using RDKit at each step. This highlights the flexibility of our zeroth-order framework to handle non-differentiable or "black-box" oracles without surrogate training.
> > > 2. **Performance:** Both GGDiff variants successfully reduce the Mean Absolute Error (MAE) compared to the unconstrained baseline, with **GGDiff-Z** achieving the lowest error rates ($\mu$: 0.5354, HOMO: 0.0835).
> > > 3. **Backbone Differences:** We note that the absolute MAE values for the unconstrained baseline differ from those reported in the DiGress paper, which are worse than the ones reported here. This is expected, as we utilize the **GDSS** backbone rather than the discrete DiGress backbone. However, the relative improvement demonstrates that our guidance mechanism effectively steers the generative process toward the target properties.
> > >
> > > |Method|$\mu \downarrow$|HOMO $\downarrow$ |
> > > |---|---|---|
> > > |Uncons.|0.9550|0.1116|
> > > |GGDiff-C|0.6248|0.1014|
> > > |GGDiff-Z|0.5354|0.0835|
> > > |DiGress-Guidance|0.81|0.56|
> > >
> > >
> > > Regarding the more comprehensive discussion of various guidance techniques, we have added a detailed examination of the different guidance techniques, including **classifier-free guidance** and **classifier-based guidance**, explicitly outlining their respective advantages and limitations. Due to current space constraints, we have placed this discussion in Appendix D for now. However, we intend to integrate this section into the main text in the final camera-ready version.

---

> > > ### Author Response · Authors · 2025-12-03
> > > **Response to reviewer KRg4 - Issues with the evaluation (2/3)**
> > >
> > > - "Most metrics are reported without standard deviations, which makes it difficult to reliably interpret performance. In the few cases where standard deviations are provided (e.g., Table 4), differences appear to be small or not statistically significant."
> > >
> > >    We thank the reviewer for this comment. Please note that, in most of the cases, the reported metrics (e.g. Val$_{\mathcal{C}}$, $\Delta$MMD) are computed over the entire population of generated graphs from a single experimental run. As these are aggregate statistics for the full sample rather than an average over multiple trials, there is no associated variance or standard deviation to report. We therefore included standard deviation metrics just where they were applicable.
> > >
> > >
> > > - "In Table 1, the reported $\Delta$MMDs obtained using your guidance methods are worse than those of the unconditional models."
> > >
> > >    We thank the reviewer for this observation. We would like to clarify that the variations in $\Delta$MMD reflect the nature of conditional generation. The $\Delta$MMD metric compares the generated graphs against the **entire test set distribution**. Since our guidance methods explicitly restrict generation to a specific conditional distribution (e.g., graphs fulfilling a constraint), we generally expect the generated distribution to deviate from the full distribution of the test set, which includes graphs that do not fulfill the constraints. This naturally results in a lower (or negative) $\Delta$MMD compared to the unconstrained model, which aims to cover the dataset's entire variance. However, we also observe cases where $\Delta$MMD is positive. This occurs when the conditional distribution aligns more closely with the high-density regions of the test set than the unconstrained baseline. In these instances, guidance not only enforces the constraint but also steers generation towards more realistic or valid parts of the distribution, effectively filtering out the out-of-distribution noise that the unconstrained model might produce.
> > >    Lastly, we would like to mention that these variations in $\Delta$MMD align with previous guidance papers, such as PRODIGY.
> > >
> > > - "The edge-minimization experiment remains unclear. As currently presented, why is the “no-edge” solution not a trivial optimum?"
> > >
> > >    We thank the reviewer for raising this point. Indeed, the empty graph is the trivial global optimum for the edge-minimization constraint itself. However, the objective of conditional generation is not merely to find a general graph that optimizes the constraint (reward), but rather to sample from the posterior distribution. This posterior is a combination of the prior (the learned data distribution) and the reward likelihood (the imposed constraint). If the guidance strength $\lambda$ is set too high, the sampler will prioritize the constraint, effectively ignoring the prior. In the case of edge-minimization, this leads to a collapse onto the trivial no-edge solution. While this solution is $100\%$ valid under the constraint, it yields a sample with very poor distributional quality, as quantified by a high MMD. This behavior is empirically demonstrated in our ablation study (Table 10), where increasing $\lambda$ (e.g., $\lambda=0.5$) achieves a $100\%$ constraint satisfaction ratio. For the main results, we intentionally selected hyperparameters that balance the high constraint satisfaction with the preservation of distributional fidelity (low MMD). This ensures the generated graphs remain structurally meaningful and representative of the underlying data distribution, preventing the collapse to the trivial solution. Consequently, this experiment serves as a valuable setting to assess the guidance mechanism's ability to minimize a cost function while preserving the structural properties of the learned prior distribution.
> > >
> > > - "No alternative guidance methods (other than Prodigy) using the same backbone are included for comparison, which limits the fairness and interpretability of the results."
> > >
> > >   We refer the reviewer to our responses to the previous comment "Comparison with baselines". Also, we would like to highlight that our evaluation is not limited to Prodigy; we explicitly compare against DiGress guidance as a key baseline. Furthermore, as detailed in the General Response, we have included GruM as a new backbone to demonstrate the framework's generality across different architectures, adding a new baseline methods alongside a new backbone.

---

> > > ### Author Response · Authors · 2025-12-03
> > > **Response to reviewer KRg4 - Guidance for Discrete Models (3/3)**
> > >
> > > ### Guidance for Discrete Models
> > >
> > > Thank you for raising this crucial point about **conditional independence**. We agree that this is the core methodological difference that allows our approach to incorporate general guidance terms.
> > >
> > >
> > > #### 1. Continuous Diffusion Models and Guidance
> > >
> > > In continuous diffusion models, conditional independence across dimensions is naturally present in the forward (noising) process. However, the reverse dynamics are governed by the score function, which gives a joint (and typically non-factorized) update over the entire state, so the reverse dynamics need not factorize over dimensions.
> > >
> > > Crucially, when guidance is incorporated via $\nabla_{x_t} \log p(y|x_t)$ (the gradient of the conditioning term), the guidance term itself also **does not need to factorize** over the individual dimensions. The model follows a non-factorized gradient that simultaneously steers the entire data sample $x_t$ towards the condition $y$. The conditionally independence is maintained in the prior noise, but the guidance term can influence the sampling globally.
> > >
> > > #### 2. The Discrete Challenge (e.g., DiGress)
> > >
> > > In discrete models (like DiGress), the reverse process involves sampling from a categorical distribution that factorizes over components (e.g., each edge $l$ in a graph): $p(G_{t-1}|G_{t}) = \prod_{l=1}^L p(G_{t-1}^l|G_{t})$.
> > >
> > > If the goal is to modify the proposal distribution directly, as in DiGress, the idea is to sample from $p(G_{t-1}|G_{t}, y) \propto p(y|G_{t-1})\prod_{l=1}^L p(G_{t-1}^l|G_{t})$. However, for the sampling process to remain edge-by-edge, the entire likelihood must factorize, meaning the **guidance term $p(y|G_{t-1})$ must be forced to factorize** as well: $p(G_{t-1}|G_{t}, y) = \prod_{l=1}^L p(y|G_{t-1}^l)p(G_{t-1}^l|G_{t})$. This necessity to factorize the guidance term is often an undesirable approximation for global graph properties.
> > >
> > > #### Our Mirror Langevin Dynamics (MLD) Approach
> > >
> > > Our proposed method leverages a continuous **relaxation** in the dual space to *decouple* the non-factorized guidance from the factorized discrete prior:
> > >
> > > 1.  **Proposal Step (Prior):** We first generate candidate logits using the factorized discrete diffusion prior, $\prod_{l=1}^L p(G_{t-1}^l|G_{t})$. This is where the **conditional independence remains**, necessary for the discrete reverse process.
> > > 2.  **Exploration Step (Guidance):** We then perform **Mirror Langevin Dynamics** (MLD) in the unconstrained logit space. This MLD update approximates sampling from the full posterior $p(G_{t-1}| y) \propto p(y|G_{t-1})p(G_{t-1})$ using:
> > >     * A continuous **Gaussian noise** (Langevin) in the logit space.
> > >     * A continuous **gradient** derived from the non-factorized guidance reward $p(y|G_{t-1})$.
> > >
> > > By operating in the continuous **dual/logit space**, the **non-factorized** reward $p(y|G_{t-1})$ can exert a steering influence on the entire graph structure **$G_{t-1}$** via the MLD steps, *without* requiring the guidance term to factorize over edges $\prod_{l=1}^L p(y|G_{t-1}^l)$. The conditional independence only applies to step 1 (proposal step), but the step 2 can use a joint update over the different dimensions.
> > >
> > > This relaxation is the key distinction. Furthermore, it allows us to robustly handle **non-differentiable rewards** $p(y|G_{t-1})$ using our Zero-Order (ZO) gradient approximation, which is less straightforward when modifying the sampling distribution directly, as is done in DiGress.
> > >
> > > Hope this clarify our proposal for extending ZO optimization to the discrete setting.
> > >
> > > [1] Du, Yilun, Conor Durkan, Robin Strudel, Joshua B. Tenenbaum, Sander Dieleman, Rob Fergus, Jascha Sohl-Dickstein, Arnaud Doucet, and Will Sussman Grathwohl. "Reduce, reuse, recycle: Compositional generation with energy-based diffusion models and MCMC." In ICML 2023.

---

### Official Review · Reviewer_vgtB · 2025-11-01

**Soundness:** 3
**Presentation:** 3
**Contribution:** 2
**Rating:** 6
**Confidence:** 3

**Summary:**

This paper proposes GGDiff, a plug-and-play framework for controllable graph generation based on stochastic optimal control (SOC).
Instead of training a new model, the authors reinterpret conditional diffusion as a control problem. They introduce a control signal Ut​ , during the sampling process to steer the trajectory toward graphs that satisfy a given reward or constraint.
The key idea is that this framework can handle both differentiable and non-differentiable rewards: for the former, they use gradient-based guidance; for the latter, they adopt zeroth-order optimization to approximate the control direction. The method builds on pre-trained graph diffusion models and does not require any retraining.

**Strengths:**

1. Novel theoretical framing
The paper reformulates conditional graph diffusion as a stochastic optimal control (SOC) problem, which provides a unified theoretical view for guided generation. This is conceptually new in the graph domain.
2. Training-free
The method only modifies the sampling process and does not require retraining or auxiliary classifiers, making it lightweight and compatible with existing pretrained diffusion models.

**Weaknesses:**

1. Lack of stability or convergence analysis
The paper does not analyze how control strength (λ) or step size affects sampling stability.
Since the control term directly modifies the diffusion dynamics, large or inconsistent Ut may cause sampling divergence, but this is not discussed.
2. High variance in zeroth-order (ZO) gradient estimation
For non-differentiable rewards, ZO estimation introduces significant stochastic noise, especially in high-dimensional graph spaces.
The method lacks mechanisms (e.g., variance reduction, adaptive sampling) to mitigate this issue.
2.No evaluation of computational efficiency
ZO optimization requires multiple reward evaluations per step, but the paper does not report runtime or sampling cost, leaving scalability unclear.
4. Narrow fairness evaluation
Only ΔDP (dyadic parity) is reported. Other fairness metrics would make the evaluation more comprehensive.

**Questions:**

1. On the choice of SOC vs. RL:
Since reinforcement learning (RL) can also optimize non-differentiable rewards, what are the concrete advantages of the SOC formulation compared to an RL-based policy optimization framework?
Did the authors attempt or consider an RL baseline for comparison?
2. Variance in zeroth-order optimization:
Zeroth-order (ZO) gradient estimation can be quite noisy in high-dimensional graph spaces.
How sensitive is GGDiff to the number of sampled directions or to the noise level?
Have the authors explored any variance reduction strategies?

---

> ### Author Response · Authors · 2025-11-21
> **Response to reviewer vgtB (1/3)**
>
> ## Lack of stability or convergence analysis
>
> We thank the reviewer for raising this critical point regarding the sensitivity of the sampling process to the guidance parameters. We respectfully direct the reviewer to Appendix E.4.1, where we have already included a detailed ablation study specifically analyzing the impact of the guidance strength ($\lambda$) on stability and convergence.
>
> As the reviewer hypothesized, our analysis confirms that large guidance terms can disrupt the diffusion dynamics. As shown in Table 10 (Appendix E.4.1):
> - Stability Region: The method is stable for $\lambda \in [0, 1]$, with optimal constraint satisfaction at $\lambda=0.5$.
> - Divergence: For $\lambda \ge 50$, the constraint satisfaction ratio drops to 0.00, explicitly confirming that excessive control strength causes the sampling process to diverge.
>
> Furthermore, we also analyze how $\lambda$ impacts distributional quality (see Table 11) via the Fréchet ChemNet Distance (FCD). Results in Table 11 show a "sweet spot" around $\lambda=0.3$, after which sample quality degrades (FCD increases) even before total divergence occurs.
>
> We have updated the main text to reference this appendix more prominently to ensure this stability analysis is easily accessible.
>
> ## Variance in ZO optimization
>
> We thank the reviewer for this important comment about the variance of the ZO estimator. To address this concern quantitatively, we have performed a new ablation study analyzing the accuracy of our gradient estimation.
>
> 1. Quantitative Error Analysis: We evaluated the relative error of the estimated gradient as a function of the number of sensing directions $N$. We defined the error as $\|\frac{\nabla r}{\|\nabla r\|_2}-\frac{\hat{\nabla} r}{\|\hat{\nabla} r\|_2}\|_2$, where $\nabla r$ is the true analytical gradient (obtained using a differentiable reward function for validation purposes) and $\hat{\nabla} r$ is the gradient estimated by our ZO approach.
> As shown in Appendix E.4.4 and in the table below, the estimation error decreases monotonically as $N$ increases. This confirms that while ZO estimation introduces variance, it can be effectively controlled simply by adjusting $N$, without requiring complex variance reduction mechanisms.
>
> |$N$|1|2|4|6|8|10|12|14|16|18|20|
> |-:|-:|-:|-:|-:|-:|-:|-:|-:|-:|-:|-:|
> |$\|\frac{\nabla r}{\|\nabla r\|_2}-\frac{\hat{\nabla} r}{\|\hat{\nabla} r\|_2}\|_2$|0.1751|0.0459|0.0028|0.0039|0.0044|0.0038|0.0037|0.0023|0.0008|0.0008|0.0001|
>
> 2. Necessity of Variance Reduction Techniques: Regarding the reviewer’s suggestion to incorporate mechanisms such as variance reduction or adaptive sampling, we demonstrate that they are not strictly necessary for our specific application, for the following reasons:
>    - Performance Scaling with $N$: Our ablation study in Table 10 (Appendix E.4.1) confirms that simple Monte Carlo averaging (increasing $N$) is sufficient to handle variance. As we increase the number of sensing directions for GGDiff-Z, the constraint satisfaction ratio improves significantly, rising from 0.172 (unguided) to 0.680 (at $N=12$). This steady improvement demonstrates that the method is robust; increasing the sample budget $N$ effectively averages out stochastic noise without requiring complex auxiliary variance reduction schemes.
>    - Competitive Downstream Performance: Despite the theoretical variance, GGDiff-Z consistently produces good results. For example, in the cycle minimization task (Table 2) and fair graph generation (Table 3), GGDiff-Z outperforms unconstrained baselines and achieves the best node-level fairness ($\Delta \text{DP}_{\text{node}}$) on the GDSS backbone.
> 3. Scope of Contribution: Our primary contribution is the unifying SOC-based framework that enables plug-and-play guidance. We implemented a standard ZO estimator to demonstrate this flexibility. Since the basic implementation yields competitive results and scales well with $N$, we opted not to increase complexity with additional variance reduction techniques, though our framework remains compatible with them.
>
> We have added the gradient error analysis to Appendix E.4.4 to provide transparency regarding the estimator's precision.

---

> ### Author Response · Authors · 2025-11-21
> **Response to reviewer vgtB (2/3)**
>
> ## Runtime and sampling cost
>
> We thank the reviewer for this comment. We agree that analyzing the computational overhead of ZO optimization is critical. We respectfully refer the reviewer to Appendix E.4.2 and the newly added Appendix E.4.3, where we provide a comprehensive evaluation of runtime and scalability.
>
> Specifically, we address computational efficiency from two perspectives:
> - Scaling with Number of Directions ($N$): In Appendix E.4.2 (Table 12), we report the total sampling time across various datasets (QM9, ZINC250k, Enzymes) while increasing the number of candidate directions $N$. We observe that the computational cost scales linearly with $N$, which is expected as each direction requires a reward function evaluation. For example, on QM9, increasing $N$ from 0 (unguided) to 5 increases sampling time from 20.24s to 68.76s. On larger graphs like ZINC250k, the time scales from 112.40s to 402.02s.
> - Scaling with Guidance Steps ($t_0$): In the newly added Appendix E.4.3, we analyze the trade-off between the duration of guidance and sampling cost. As shown in the ablation study, applying guidance only during the later stages of diffusion significantly reduces runtime (e.g., decreasing from 226s at full guidance to 96s at 10% guidance), though this comes with the performance trade-offs discussed in the previous responses.
>
> We believe these results demonstrate that while ZO optimization introduces overhead, the cost is predictable (linear, both in terms of $N$ and $t_0$) and manageable for standard molecular and graph generation benchmarks.
>
> ## Other fairness metrics
>
> We thank the reviewer for this comment. We would like to clarify the specific scope of "fairness" addressed in this work.
>
> Our objective is to ensure structural fairness in the generated topology—specifically, ensuring that the distribution of edges between nodes of different sensitive groups satisfies statistical parity (Dyadic Parity). We do not focus on downstream algorithmic fairness (e.g., GNN classification parity), but rather on the intrinsic fairness of the graph structure itself.
>
> To the best of our knowledge, the standard metrics for evaluating this structural fairness are those proposed in the recent literature by Navarro et al. (NeurIPS 2024). Following their methodology, we reported both:
>
> - $\Delta$DP: The global Dyadic Parity difference.
> - $\Delta$DP$_{\text{node}}$: The node-level Dyadic Parity difference (which addresses local fairness).
>
> We conducted a literature search for alternative metrics specifically designed for generative graph topology but found that $\Delta$DP remains the primary standard for this task. Nevertheless, we are actively investigating and analyzing other potential metrics that could further substantiate the fairness of our approach and, if the reviewer has specific alternative metrics or references in mind that are applicable to structural graph generation, we would be more than happy to incorporate them into our evaluation.

---

> ### Author Response · Authors · 2025-11-21
> **Response to reviewer vgtB (3/3)**
>
> ## SOC vs. RL
>
> We thank the reviewer for this important question. The reviewer is correct that both SOC and RL can optimize non-differentiable rewards, and we appreciate the opportunity to clarify the relationship and our methodological choice. We have included a new Appendix F with the contents of this response, which are repeated below.
>
> ### Relationship between SOC and RL
>
> The SOC and RL formulations for diffusion models share the same underlying mathematical formulation: both can be viewed as solving a Markov Decision Process over the diffusion trajectory. In fact, our approach is closely related to the framework of *value-weighted sampling* described by Uehara et al. (2024) (Section 6.2) [2].
>
> However, when optimizing the policy (the transition kernel, and therefore, the score function), they differ fundamentally in their **goal**:
>
> ### RL-based approaches (e.g., DPOK [3], RLDF [4]):
>
> - **Goal**: Fine-tune/optimize the diffusion model parameters $\theta$ (the transition kernel) to maximize expected reward;
> - **Requirements**: Access to training data and the ability to retrain the model;
> - **Advantage**: Modify the model's distribution to better align with the reward;
> - **Limitation**: Requires retraining for each new conditioning task; needs sufficient data for policy optimization.
>
> ### SOC (our method):
>
> - **Goal**: Design an optimal controller $\mathbf{U}(\mathbf{G}_t^c, t)$ that guides a frozen pre-trained diffusion model
> - **Requirements**: Only a pre-trained unconditional model and a reward/likelihood function
> - **Advantage**: Plug-and-play at inference time; single model handles multiple tasks; no retraining needed
> - **Limitation**: Constrained to modify the base process rather than fully optimizing it
>
> ### Connection to Value-Weighted Sampling
>
> Uehara et al. demonstrate guidance can be posed as a value-weighted sampling procedure, with value function given by
>
> $$V^*(x_t, t) = \mathbb{E}_{x_0 \sim p(x_0 | x_t)} [\exp(r(x_0))]$$
>
> and the optimal control is given by the gradient of $\log V^*(x_t, t)$.
> This is exactly the expression that we obtained, adapted to the discrete graph setting. Hence, value-weighted sampling with RL and SOC are equivalent.
> We took the SOC path because we believed is a more principled way of seeing this problem of controlling the trajectory of the diffusion model.
>
> On the other hand, the reason why we use SOC rather than fine-tuning or policy optimization lies in the following facts:
>
> 1. **No task-specific training data**: In applications like fairness-aware graph generation or optimizing novel graph properties, we often lack the labeled datasets required for RL-based fine-tuning.
>
> 2. **Multi-task flexibility**: A single unconditional model can be steered toward different objectives at inference time by simply changing the reward function $r(\cdot)$, without any retraining.
>
> 3. **Inference-time control**: The SOC formulation provides an analytical solution (via the HJB equation and Feynman-Kac formula) for the optimal control:
>
> $$
> \\mathbf{U}^\*(\\mathbf{G}\_t^c, t) = -g(t)\\nabla\_{\\mathbf{G}\_t^c} \\log \\mathbb{E}^{\\text{prior}}\\left[\\exp\\left(\\frac{-r(\\mathbf{G}\_0^c)}{\\lambda}\\right) \\Big| \\mathbf{G}\_t^c\\right]
> $$
>
> This reveals that optimal guidance reduces to learning a value function, which can be done efficiently without access to training data.
>
> 4. **Practical efficiency**: For practitioners, our method means: train one unconditional model once, then apply it to arbitrary downstream tasks by specifying reward functions—no dataset collection or model retraining required.
>
> In a nutshell, our SOC formulation is not a replacement for RL-based methods, but rather addresses a complementary and practically important setting: **conditional generation when task-specific training data is unavailable**. The key contribution is enabling a single pre-trained unconditional model to be steered toward arbitrary objectives at inference time, without retraining.
>
> [1] Jo, J., Kim, D., & Hwang, S. J. "Graph generation with diffusion mixture". ICML 2024
>
> [2] Uehara et al. (2024). "Fine-Tuning of Continuous-Time Diffusion Models as Entropy-Regularized Control", Section 6.2
>
> [3] Fan, Ying, Olivia Watkins, Yuqing Du, Hao Liu, Moonkyung Ryu, Craig Boutilier, Pieter Abbeel, Mohammad Ghavamzadeh, Kangwook Lee, and Kimin Lee. "DPOK: Reinforcement learning for fine-tuning text-to-image diffusion models." in NeurIPS, 2023.
>
> [4] Zhao, Yulai, Masatoshi Uehara, Gabriele Scalia, Sunyuan Kung, Tommaso Biancalani, Sergey Levine, and Ehsan Hajiramezanali. "Adding Conditional Control to Diffusion Models with Reinforcement Learning." In ICLR 2025.

---

### Author Response · Authors · 2025-12-03
**Summary of the changes**

We thank all reviewers for their constructive feedback. Addressing these comments has significantly strengthened our work. Before the end of the rebuttal period, we summarize below the main contributions and the improvements made in response to the reviews.

## Additional experiments and baselines

1. **Expanded ablations and analysis.**
We added new ablations on our method, including: a convergence study w.r.t. $\lambda$; an analysis of the variance of the ZO estimator and the running time (Reviewer vgtB); a study of sample efficiency with and without guidance; and an ablation where guidance is applied only at specific timesteps (Reviewer cZPL). We also include a diversity–quality trade-off analysis (Reviewer KRg4).

2. **Generality across backbones (GruM).**
To demonstrate the plug-and-play nature of our framework, we implemented GruM as a new backbone. We show that GGDiff effectively guides this state-of-the-art model, achieving improved results in fairness and incomplete graph generation tasks compared to the GDSS-based version (Reviewer KRg4).

3. **Comparison with DiGress and black-box rewards.**
We performed a new experiment on conditional molecular generation (targeting Dipole moment and HOMO) to compare explicitly against DiGress and using RDKit directly as a black-box reward (Reviewers cZPL and KRg4).

4. **Clarified metrics and results.**
We improved the description of evaluation metrics, such as the edge-minimization setup, and refined the presentation of experimental results (Reviewer KRg4).

5. **Improved discussion of non-differentiable rewards.**
We clarified the strengths of our method for handling non-differentiable rules/rewards. The new experiment using rdkit as a black-box reward demonstrates these advantages explicitly (Reviewers cZPL and KRg4).

## Additional discussion and clarifications

1. **Broader discussion of guidance methods.**
- We added a comparison of our SOC framework with RL and value-based sampling approaches (Reviewer vgtB).
- We expanded the discussion contrasting our method with classifier-based and classifier-free guidance (Reviewer KRg4).

2. **Extension to discrete domains.**
Following Reviewer KRg4’s suggestion, we included a concrete extension of our framework to discrete settings.

3. **Clarification on fairness experiments.**
We elaborated on the novelty, motivation, and challenges of our fairness-constrained sampling task, clarifying the intended scope of these experiments (Reviewer vgtB).

---

### Meta-Review · Area_Chair_wECk · 2025-12-26

**Summary:**

The paper proposes Graph Guided Diffusion (GGDiff), a framework that reformulates conditional graph generation as a stochastic optimal control (SOC) problem. The authors aim to provide a *plug-and-play* guidance mechanism for pre-trained continuous diffusion models, that can handle both differentiable and non-differentiable reward functions at inference time.

While the approach is technically sound, the paper raises several concerns that limits its suitability for acceptance. In particular, the empirical validation remains limited, comparisons to existing guidance methods are incomplete, and the demonstrated scope of the method is narrower than what is suggested by the paper’s claims. Overall, despite some technical merit, the work does not provide sufficiently strong or convincing evidence to meet the bar for acceptance.

**Reviewer Concerns:**

**Addressed concerns**
- To address concerns about reliance on a single backbone (GDSS), the authors implemented and evaluated their method on an additional backbone (GruM), which partially supports the claim of plug-and-play applicability.
- Concerns regarding high variance and computational cost were addressed through additional ablation studies.

**Outstanding concerns**
- While the paper emphasizes generality, the proposed framework is limited to continuous diffusion models. Much of the recent progress in graph generation has shifted toward discrete diffusion formulations. While the rebuttal discusses a possible extension to the discrete setting, this remains speculative and unvalidated
- Reviewers raised several issues about the evaluation, many of which remain only partially resolved. For example, 1) most metrics are computed from a single run without reporting standard deviations, making comparisons and interpretations difficult. 2) The $\Delta\text{MMD}$ metric is extensively used in the work, but it is hard to interpret in the context of conditional generation. Moreover, the kernel associated with the MMD was not even specified in the paper. These issues make it difficult to assess the true impact and soundness of the empirical results.
- Despite additional comparisons to DiGress, the experimental evaluation still lacks thorough comparisons against strong and relevant guidance methods such as CFG, surrogate-based guidance, and other conditional generation approaches. While the authors provide conceptual arguments for the advantages of GGDiff, the current experiments are still not strong enough to substantiate all the claims.

**Reviewer Scores:**

I expect both Reviewer KRg4 and cZPL to maintain their original negative scores. Specifically,
- Reviewer KRg4's concerns about narrow scope, insufficient baseline comparisons and evaluation remain largely unaddressed.
- Reviewer cZPL's concerns about insufficient baseline comparisons also remain outstanding.

---

### Decision · Program_Chairs · 2026-01-26

Reject